



# Comparison of root water uptake models in simulating $CO_2$ and $H_2O$ fluxes and growth of wheat

Thuy Huu Nguyen[1], Matthias Langensiepen[1], Jan Vanderborght[3], Hubert Hüging[1], Cho Miltin Mboh[1], Frank Ewert[1, 2]

[1]University of Bonn, Institute of Crop Science and Resource Conservation (INRES), Katzenburgweg 5, 53115 Bonn, Germany

[2]Leibniz Centre for Agricultural Landscape Research (ZALF), Institute of Landscape Systems Analysis, Eberswalder Strasse 84, 15374 Muencheberg, Germany

[3]Agrosphere, Institute of Bio- and Geosciences (IBG-3), Forschungszentrum Jülich GmbH, 52428, Jülich, Germany

*Correspondence to*: Thuy Huu Nguyen (tngu@uni-bonn.de)

**Abstract.** Stomatal regulation and whole plant hydraulic signaling affect water fluxes and stress in plants. Land surface models and crop models use a coupled photosynthesis-stomatal conductance modelling approach and estimate the effect of soil water stress on stomatal conductance directly from soil water content or matrix potential without explicit representation of hydraulic signals between the soil and stomata. In order to explicitly represent stomatal regulation by soil water status as a function of the hydraulic signal and its relation to the whole plant hydraulic conductance, we coupled the crop model LINTULCC2 and

the root growth model SLIMROOT with Couvreur's root water uptake model (RWU), and the HILLFLOW soil water balance model. Since plant hydraulic conductance depends on the plant development, this model coupling represents a two-way coupling between growth and plant hydraulics. To evaluate the advantage of considering plant hydraulic conductance and hydraulic signaling, we compared the performance of this newly coupled model with another commonly used approach that relates root water uptake and plant stress directly to the root zone water potential (HILLFLOW with Feddes' RWU model).

Simulations were compared with gas flux measurements and crop growth data from a wheat crop grown under three water supply regimes (sheltered, rain-fed and irrigated) and two soil types (stony and silty) in Western Germany in 2016. The two models showed a relatively similar performance in simulation of dry matter, LAI, root growth, RWU, gross assimilation rate, and soil water content. The Feddes model predicts more stress and less growth in the silty soil than in the stony soil, which is opposite to the observed growth. The Couvreur model better represents the difference in growth between the two soils and the

different treatments. The newly coupled model (LINTULCC2 - SLIMROOT - Couvreur - HILLFLOW) was also able to simulate the dynamics and magnitude of whole plant hydraulic conductance over the growing season. This demonstrates the importance of two-way feedbacks between growth and root water uptake for predicting the crop response to different soil water conditions in different soils. Our results suggest that a better representation of the effects of soil characteristics on root growth is needed for reliable estimations of root hydraulic conductance and gas fluxes, particularly in heterogeneous fields.

The newly coupled soil-plant model marks a promising approach but requires further testing for other scenarios regarding crop, soil, and climate.



# 1 Introduction

Soil water status is amongst the key factors that influence photosynthesis, evapotranspiration and growth processes (Hsiao, 1973). Accurate estimation of crop water stress responses is important for predictions of crop growth, yield, and water use by

crop models and land surface models (Egea et al., 2011).

Crop models and land surface models lump the effects of soil water deficit on stomatal regulation and crop growth in so-called 'stress factors' (Verhoef and Egea, 2014; Mahfouf et al., 1996). Crop water stress is strongly influenced by soil water availability which in turn depends on the distribution of water and of roots in the root zone and the transpiration rate or total root water uptake. Adequate representations in simulation models of root water uptake (RWU) and root distributions (Gayler

et al., 2013; Wöhling et al., 2013; Zeng et al., 1998; Desborough, 1997) are therefore needed. Most macroscopic RWU models estimate the water uptake as a function potential transpiration (i.e. the transpiration of the crop when water is not limiting) and average moisture content or water potential and rooting densities (Feddes et al., 2001; van Dam, 2000). However, in this representation of RWU, crucial relations between RWU model parameters and root and plant hydraulic conductances, which translate soil water potentials to water potentials in the shoot to which stomata respond, were lost. For instance, the water stress

factor calculated by the Feddes model (Feddes et al., 1978) based on the soil water potentials involves indirect linkages between the root zone water potential and the water potential in the shoot in the sense that the water stress factors are adapted when potential transpiration rate changes. Such models like the Feddes approach represent in an indirect manner the role of the root and plant hydraulic conductance and thus require calibration for different crop types and growing seasons (Cai et al., 2018; Vandoorne et al., 2012; Wesseling et al., 1991). The conductance of the root system is an important feature of the root system

and different approaches to include it in models of root water uptake were published (Quijano and Kumar, 2015; Vadez, 2014, Kramer and Boyer, 1995; Peterson and Steudle, 1993). Plant hydraulic conductance determines leaf water potentials which have a significant impact on stomatal conductance, leaf gas exchange, and leaf growth (Tardieu et al., 2014; Trillo and Fernández, 2005; Sperry, 2000; Zhao et al., 2005; Gallardo et al., 1996). Recently, some one-dimensional macroscopic RWU models based on hydraulic principles have been developed to represent water potential gradients from soil to root (de Jong van

Lier et al., 2008) and within the root system (Couvreur et al., 2014). The latter approach simplified a physically based description of water flow in the coupled soil-root system accounting for the root system hydraulic properties and architecture to simple linear equations between soil water potentials, leaf water potential, root water uptake profiles and transpiration rate that can be solved directly. It thereby avoids computation time consuming numerical solutions of ordinary differential equations for the water flow and balance in the root system that are coupled with the non-linear soil water balance partial

differential equation. It uses a stomatal regulation model assuming that stomatal conductance is not influenced by the leaf water potential as long as the leaf water potential is above a critical potential threshold. Leaf water potential is kept constant by changing stomatal conductance when the critical leaf water potential threshold is reached. The Couvreur model also allows presenting the different stomatal regulations [i.e. isohydric and anisohydric in Tardieu and Simonneau, (1998)] (Couvreur et al., 2014, 2012).



Recently, inverse modelling routines using datasets of root density, shoot leaf area, and soil water content and potential permitted the quantification of root-related parameters of Couvreur's model (root hydraulic conductivity). Sap flow measurements were used to validate simulated RWU by the parameterized model (Cai et al., 2018; Cai et al., 2017). These studies demonstrated the close relation between the root system conductance and root growth as part of overall plant growth and its response to water stress pointing at a two-way coupling between root-water uptake and plant growth. This implies that

the parameterization of root water uptake needs to be coupled to plant growth, which in turn is influenced by water stress and other factors. Plant hydraulic conductance was introduced in crop models for several field crops such as soybean (Olioso et al., 1996), winter wheat (Wang et al., 2007), or for model testing (Tuzet et al., 2003). However, plant hydraulic conductance in these studies was kept constant without reference to dynamic root growth. To our knowledge, the effect of a two-way coupling between a RWU model accounting for whole plant hydraulic regulation and a crop growth model has not been studied

yet. It is unclear whether such a coupled model improves the simulation of crop growth and development, $CO_2$ and $H_2O$ fluxes. In this study, we coupled the Couvreur's RWU model (Couvreur et al., 2014; Couvreur et al., 2012) with the existing crop growth model LINTULCC2 (Rodriguez et al., 2001) to consider the whole plant hydraulic conductance from root to shoot. The dynamics of root and shoot growth under varying soil water availability are explicitly represented by the coupled model. The overall aim of the study was to investigate whether consideration of plant hydraulic conductance can improve the

simulation of $CO_2$ and $H_2O$ fluxes, and crop growth in biomass, roots, and leaf area index of the same crop that is grown in two different soils and for three different water application regimes. To achieve this aim, three objectives were addressed: (i) analyse and compare the predictive quality of a crop growth model coupled with a RWU model that considers plant hydraulics (Couvreur RWU model) and a model that does not consider plant hydraulics (Feddes RWU model), (ii) compare the simulated plant hydraulic conductances for the different growing conditions with direct estimates of these conductances from

measurements, and (iii) analyse the sensitivity of RWU and crop growth to the Couvreur RWU and root growth model parameters (root hydraulic conductance, critical leaf water potential threshold, and specific weight of seminal and lateral root)

## 2 Materials and Methods

### 2.1 Location and experimental set-up

The study area was located in Selhausen in North Rhine-Westphalia, Germany (50°52'N, 6°27'E). The study field is slightly

inclined with a slope of around 4° and characterized by a strong gradient in stone content along the slope (Stadler et al., 2015). Two rhizotrones were set up in the field: the upper site with stony soil (hereby F1) contains up to 60% gravel by weight while in the lower site with silty soil (hereby F2) the gravel content was approximately 4%. At each study site the effects of three different water treatments on growth and fluxes were investigated (sheltered – P1, rainfed – P2, and irrigated – P3) (Fig. 1). Further information on the field experiment and set-up are presented in Cai et al., (2016), Stadler et al., (2015), and Cai et al.,

(2018). Irrigation was applied two times: on 22 May and 26 May 2016 in the irrigated plots (F1P3 and F2P3) during the growing season using dripper lines. The dripper lines (Model T-Tape 510-20-500, Wurzelwasser GbR, Münzenberg, Germany)



were installed with 0.3-m intervals and parallel to crop rows. The non-transparent plastic shelter was manually covered (11 times) during rainfall and removed when rain stopped to induce water stress. On the sheltered days, radiation was assumed to be zero for the sheltered plots. Winter wheat (*Triticum aestivum* cv. Ambello) was sown with a density of 350-370 seed m$^{-2}$

on 26 October 2015 and harvested on 26 July 2016 in both the stony (F1) and silty (F2) parts of the field. Fertilizers were applied at a rate of 80 kg N + 60 kg K$_2$O + 30 kg P$_2$O$_5$ per hectare on 15 March 2016. Nitrogen was further added on 2 May and 7 June 2016 with 60 and 50 kg N per hectare, respectively. Weeds and pests were controlled according to standard agronomic practice.

[Insert Fig. 1 here]

**2.2 Measurements**

**2.2.1 Soil water measurement, soil property, and root growth**

Soil water content and soil water potential were measured hourly by home-made time domain reflectometer (TDR) probes (Cai et al., 2016), tensiometers (T4e, UMS GmbH), and dielectric water potential sensors (MPS-2 matric potential and temperature sensor, Decagon Devices), respectively. Sensors were installed at 10, 20, 40, 60, 80 and 120 cm depth. Root

measurements were taken with a digital camera (Bartz Technology Corporation) repeatedly from both left and right side at 20 locations along 7 m-long horizontally installed rhizotubes (clear acrylic glass tubes with outer and inner diameters of 64 and 56 mm, respectively). The calibration of the sensors, root growth observation, and post processing of the data was described in detail in Cai et al., (2016) and Cai et al., (2017).

**2.2.2 Sap flow, leaf water potential and gas fluxes measurement**

Five, three, and five sap flow sensors (SAG3) (Dynamax Inc., Houston, USA) were installed in the irrigated, rain-fed and sheltered treatments, respectively, at the beginning of wheat anthesis when stem diameters ranged between 3-5 mm. Vertical and horizontal temperature gradients, (dT) of each sensor were recorded at 10 minute intervals with a CR1000 data logger and two AM 16/32 multiplexers (Campbell Scientific, Logan, Utah). Sensor heat inputs were controlled by voltage regulators controlled by the CR1000 data logger. The raw signal data was aggregated to 30 minutes intervals and sap flow calculated

following Langensiepen et al., (2014). The number of tillers per square meter was counted every two weeks during the operation period of sap flow sensors (26 May – 23 July 2016). Tiller numbers were used to upscale the sap flow of single tiller (g h$^{-1}$) to canopy transpiration rate (mm h$^{-1}$ or mm d$^{-1}$).

Leaf water potential was measured every two weeks from 7 am to 8pm under clear and sunny conditions from tillering (20 April) to the beginning of maturation (29 June 2016). Three to four upmost fully developed leaves from three to four different

plants were detached by a sharp knife to measure leaf water potential with a digital pressure chamber (SKPM 140/ (40-50-80), Skye Instrument Ltd, UK).



Plant hydraulic conductance in crop species can be estimated by measuring the transpiration and the root zone and leaf water potentials (Tsuda and Tyree, 2000). In our study, we calculated the conductance according to Ohm's law by dividing the hourly sap flow by the difference between root-zone water potential and leaf water potential. The root zone water potential was

calculated based on hourly measured soil water potential and measured root length density (cm cm$^{-2}$) at 6 depths (10, 20, 40, 60, 80, and 120 cm) in the soil profile following Eqs. (8) and (10) (see Section 2.3.4). During one measurement day, 6 hourly values of the conductance were obtained from measurements between 11 AM to 4 PM. The average and standard deviation of these hourly measurements were calculated for each measurement day. Yet, the hydraulic conductance can vary within short time periods due to the role of aquaporins (Maurel et al., 2008; Javot and Maurel, 2002; Henzler et al., 1999) or ABA regulation

(Parent et al., 2009), and xylem cavitation (Sperry et al., 1998). We assumed however a constant plant hydraulic conductance during the day.

Canopy gas exchange was measured hourly on the same days when leaf water potentials were measured with a closed chamber system (Langensiepen et al., 2012). $CO_2$ concentration was derived with a regression approach by Langensiepen et al., (2012). Because we were interested in comparing measured with calculated hourly instantaneous gross assimilation by the newly

coupled root: shoot model (LINTULCC2 with other subroutines), the total soil respiration (i.e. heterotrophic organisms and root respiration) was subtracted from the instantaneous canopy $CO_2$ exchange rate measured by the closed chamber. The total soil respiration was calculated based on measured soil temperature, soil water content at 10 cm soil depth, and leaf area index from crop using the fitted parameters derived from the same field and soil types (Prolingheuer et al., 2010). The calculated total soil respiration was compared and validated with the measured values in the same field in the previous years from Stadler

et al., (2015).

### 2.2.3 Crop growth

Crop growth information was collected bi-weekly from 20 April until harvest 26 July 2016. Leaf area index and crop biomass were measured by harvests of two rows (1 m each) for each treatment. Leaves were separated into green leaves and brown leaves, and the brown and green leaf area was measured using a leaf area meter (LI-3100C, Licor Biosciences, and Lincoln,

Nebraska, USA). The above ground biomass was measured using the oven drying method. Samples were first weighed in total, then separated into different plant organs (green leaf, brown leaf, stem, ear, and grain) and weighed. Subsamples were afterward extracted from these samples, weighed, dried in an oven at 105 °C for 48 hours and weighed again for determining dry matter. At the end of growing season, four replicates of one square meter of plants were harvested from the plots to determine grain yield and harvest index.



## 2.3 Model description

### 2.3.1 Description of the original LintulCC crop model

We used the crop model LINTULCC2 (Rodriguez et al., 2001). LINTULCC2 couples photosynthesis to stomatal conductance and can perform a detailed calculation of leaf energy balances (Rodriguez et al., 2001) (see Appendix A). This model was validated and compared with different crop models for spring wheat and used to simulate the effects of elevated $CO_2$ and drought conditions (Ewert et al., 2002; Rodriguez et al., 2001). LINTULCC2 calculates phenology, leaf growth, assimilate partitioning, and root growth following the procedure outline in Rodriguez et al., (2001).

In LINTULCC2, the assimilation rate of the sunlit and shaded leaf is calculated using the biochemical model of (Farquhar and Caemmerer, 1982). Stomatal conductance ($g_s$) was calculated according to the model of Leuning (Leuning, 1995) for sunlit and shaded leaves separately. In LINTULCC2 $CO_2$ uptake is calculated as a function of $CO_2$ demand by photosynthesis, and the ambient concentration of $CO_2$, using the iterative methodology proposed by Leuning (1995) (Appendix A). For, the sake of simplification, in LINTULCC2, the internal leaf $CO_2$ concentration, $C_i$, is initially assumed as 0.7 times the atmospheric $CO_2$ concentration Ca (Vico and Porporato, 2008; Rodriguez et al., (2001); Jones, 1992). Then, the light saturated photosynthetic rate of sunlit and shaded leaves (AMAXsun, and AMAXshade, $\mu M$ $CO_2$ $m^{-2}$ $s^{-1}$), and the quantum yield for sunlit and shaded leaves (EFFsun, and EFFshade, $\mu M$ $CO_2$ $MJ^{-1}$), are calculated iteratively (Farquhar et al., 1980; Farquhar, 1982). This iterative loop ends when the difference in calculated internal $CO_2$ mole fraction between two consecutive loops is $< 0.1$ $\mu mol$ $mol^{-1}$ (Appendix A). Based on a fraction of sunlit (and shaded) leaf area and LAI, the leaf stomatal resistance of sunlit and shaded leaves was integrated over the canopy leaf area to the canopy resistance ($r_s$) (Appendix B).

The canopy resistance, crop height, and calculated crop albedo (depending on both crop and soil water content of the surface layer) and the surface energy balance were used to calculate potential crop evapotranspiration (ETP – mm $h^{-1}$) using the Penman-Monteith equation (Allen et al., 1998) (see Appendix B). The obtained potential surface evapotranspiration is then split into evaporation and potential transpiration using:

$$T_{pot} = ETP\left(1 - e^{-kLAI}\right) \qquad (1)$$

where k is the light extinction coefficient [0.6 in this study (Faria et al., 1994; Mo and Liu, 2001; Rodriguez et al., 2001)].

$T_{pot}$ (mm $h^{-1}$) represents by definition the transpiration of the crop that is not limited by the root zone water potential. In section 2.3.4 it is explained how the actual transpiration, $T_{plant}$ (mm $h^{-1}$), is calculated as a function of the potential transpiration and the root zone soil water potential. The ratio $T_{plant}$ /$T_{pot}$ defines the water stress factor $f_{wat}$, which is used in the photosynthesis model:



$$fwat = \frac{T_{plant}}{T_{pot}} \qquad (2)$$

Originally, LINTULCC2 runs at daily time steps (which allows for the within day variations in temperature, radiation and vapor pressure deficit). LINTULCC2 requires daily maximum and minimum temperature, actual vapor pressure, rainfall, wind speed, and global radiation. In order to capture the diurnal response of stomata, we modified the time step of the photosynthesis

and stomatal conductance subroutine from daily to hourly, while daily time steps were kept in the remaining subroutines (phenology, leaf growth, biomass partition).

### 2.3.2 Root growth model

Root growth was simulated using SLIMROOT (Addiscott and Whitmore, 1991). The vertical extension of the seminal roots and the distribution of the lateral roots within the soil profile depend on the root biomass, the soil bulk density, the soil water

content calculated by Hillflow1D (Bronstert and Plate, 1997), and the soil temperature computed by STMPsim (Williams and Izaurralde, 2005). The supply of assimilates from the shoot (RWTR) (g m$^{-2}$ d$^{-1}$) is given by a partitioning table based on the thermal time (van Laar et al., 1997)) that is used to calculate the vertical penetration of seminal and lateral roots. The assimilate allocation for seminal root growth (ASROOT) is constrained by daily supply of assimilates from the shoot RWRT (g m$^{-2}$ d$^{-1}$) and the demand of assimilates from seminal roots (ASROOT$_{demand}$).

$$ASROOT = \min(ASROOT_{demand}, RWRT) \qquad (3)$$

ASROOT$_{demand}$ is a function of the number of seminal roots per square meter (NSROOT) which depends on the number of emerged plants per square meter and the number of seminal roots per plant; the specific weight of seminal root WSROOT (g m$^{-1}$); and the daily elongation rate of seminal roots RSROOT (m d$^{-1}$):

$$ASROOT_{demand} = RSROOT * WSROOT * NSROOT \qquad (4)$$

RSROOT depends on the soil temperature and is constrained by a maximal elongation rate, RSROOT$_{max}$ and the soil temperature depend rate which is an empirical function of the soil temperature of the deepest layer where roots are growing,

TBOTLAYER (K) (Jamieson and Ewert, 1999):

$$RSROOT = min(RSROOT_{max}, TBOTLAYER * RTFAC) \qquad (5)$$

where RTFAC is the temperature factor driving the penetration of seminal roots (m K$^{-1}$ d$^{-1}$) and TBOTLAYER (K) the soil temperature of the deepest layer where roots are growing. When soil temperature is below or equal to 0$^{o}$C, no seminal growth occurs. The maximum daily elongation rate of seminal roots, $RSROOT_{max}$ was set at 0.03 m d$^{-1}$ for wheat according to Watt et al., (2006).

The daily increment in seminal root length (SRLIR - m m$^{-2}$ d$^{-1}$) is defined as:

$$SRLIR = ASROOT / WSROOT \qquad (6)$$



Lateral roots are simulated when the root biomass supplied by the shoot is greater than the assimilate demand of seminal roots (RWRT > ASROOT$_{demand}$). Lateral root biomass is distributed stepwise from the top layer to the deepest soil layer with seminal roots.

Roots start to die after anthesis. Since the specific weight of the roots of cereal crops varies with soil strength (Colombi et al.,
2017; Lipiec et al., 2016; Hernandez-Ramirez et al., 2014; Merotto Jr and Mundstock, 1999), we chose different specific weights for the stony (F1) and silty soil (F2) from the range that was observed by Noordwijk and Brouwer (1991) and Jamieson and Ewert (1999) in soils with different soil strength (Appendix C).

### 2.3.3 Physically based soil water balance model

HILLFLOW 1D was chosen for calculating the water potentials in the soil and how they change with depth and time as a
function of the precipitation, soil evaporation, RWU, and water percolation at the bottom of the simulated soil profile (Bronstert and Plate, 1997). HILLFLOW 1D calculates soil water content and water fluxes by numerically solving the Darcy equation for unsaturated water flow in porous media (Bronstert and Plate, 1997). The relations between soil water matric potential head, water content and hydraulic conductivity are described by the Mualem-van Genuchten functions (van Genuchten, 1980). The parameters of these functions, i.e. the soil hydraulic parameters, for the different soil layers and the two sites were taken from
(Cai et al., 2018) (Appendix D). In this study, a soil depth of 1.5 m vertically discretized into 50 layers was considered. A free drainage bottom boundary and a mixed flux-matric potential boundary at the soil surface were implemented. The mixed upper boundary condition prescribes the flux at the soil surface by the precipitation and evaporation rates as long as the matric potential heads are not above or below critical heads. When these heads are reached, the boundary conditions are switched to constant matric potential boundary conditions.

**2.3.4 Feddes' and Couvreur's root water uptake models**

The Feddes RWU model (Feddes et al., 1978) (See Appendix E) was already built in the HILLFLOW 1D model (Bronstert and Plate, 1997). We implemented the Couvreur RWU model (Couvreur et al., 2014a; Couvreur et al., 2012) into HILLFLOW. Both models, T$_{plant}$ is calculated for each model from the sum of the simulated RWU in the different soil layers and used to calculate the water stress factor (fwat) following Eq. (2), which was used in the photosynthesis model. In the Feddes model,
root water uptake from a soil layer is proportional to the normalized root density, NRLD (m$^{-1}$), in that layer and is multiplied by a stress function $\alpha$ that depends on the matric potential head, $\psi_m$ (m), in that soil layer and the potential transpiration rate (see Appendix E for the definition of $\alpha$):

$$RWU_i = \alpha\left(\psi_{m,i}, T_{pot}\right) T_{pot} NRLD_i \Delta z_i \qquad (7)$$

where NRLDi is calculated from the root length density, RLD (m m$^{-3}$) and discretized soil depth $\Delta z_i$ (m) as:



$$NRLD_i = RLD_i / \sum_{i=1}^{N} RLD_i \Delta z_i \qquad (8)$$

The parameters of the α stress functions model were taken from (Cai et al., 2018) (See Appendix C). According to Eq. (7), the
reduction of water uptake in a certain layer depends on the matric potential head in that layer only and does not influence the
water uptake in other layers. This means that a reduced water uptake in dried out soil layers directly leads to a reduction of the
total root water uptake and plant transpiration and is not compensated by increased uptake in other layers where there is still
sufficient water available.

In the Couvreur model, which is based on a mechanistic description of water flow in the coupled soil-plant system, the root
water uptake in a certain soil layer is related to the water potentials in the root system and root water uptake in other soil layers
so that compensatory uptake is considered in this model. Root water uptake in a certain layer is obtained from:

$$RWU_i = T_{plant} NRLD_i \Delta z_i + K_{comp}(\psi_i - \psi_{sr}) NRLD_i \Delta z_i \qquad (9)$$

where $\psi_i$ (m) is the total water potential head (or hydraulic head which is the sum of the matric and gravitation potential heads)
in layer i, $\psi_{sr}$ (m) is the average hydraulic head in the root zone and $K_{comp}$ (d$^{-1}$) is the root system conductance for compensatory
uptake. The first term of Eq. (9) represents the uptake from that soil layer when the hydraulic head is uniform in the root zone
and the second term represents the increase or decrease of uptake from the soil layer due to a respectively higher and lower
hydraulic head in layer i than the average hydraulic head. The average root zone hydraulic head is calculated as the weighed
average of the hydraulic heads in the different soil layers as:

$$\psi_{sr} = \sum_{i=1}^{N} \psi_i NRLD_i \Delta z_i \qquad (10)$$

The plant transpiration rate is the minimum of the potential transpiration rate and the transpiration rate, $T_{threshold}$ (mm h$^{-1}$), when
the hydraulic head in the leaves reaches a threshold value, $\psi_{threshold}$ (m) that triggers stomatal closure:

$$T_{plant} = \max(0, \min(T_{pot}, T_{threshold})) \qquad (11)$$

$T_{threshold}$ is calculated from the difference between the root zone hydraulic head and the threshold hydraulic head in the leaves
$\psi_{threshold}$ that is multiplied by the plant hydraulic conductance, $K_{plant}$ as:

$$T_{threshold} = K_{plant}(\psi_{sr} - \psi_{threshold}) \qquad (12)$$



In our study, we used the a critical leaf hydraulic head, $\psi_{threshold}$ of $-200$ m (equivalent to $-2$ MPa) (Cochard, 2002; Tardieu and Simonneau, 1998). The original Couvreur model only considers the hydraulic conductance from the roots to the plant collar, $K_{rs}$, by assuming that the hydraulic resistance from plant collar to leaves is minor as compared to root system resistance.

The shoot hydraulic resistance could be large in some crop plants (Gallardo et al., 1996) or in trees (Domec and Pruyn, 2008; Tsuda and Tyree, 1997). In order to simulate the leaf water potential, the whole plant hydraulic conductance ($K_{plant}$) needs to be used. The whole plant hydraulic conductance could be estimated from different components (i.e. soil to root, stem to leaf) following an approach from Saliendra et al., (1995) or a more complex attempt by Janott et al., (2011). Because hydraulic data from plant collar to leaf are rare and difficult to obtain and account for differing species characteristics and environmental

conditions, for the sake of simplification, we derived $K_{plant}$ ($d^{-1}$) from the root hydraulic conductance ($K_{rs,doy}$) assuming that $K_{plant}$ is a constant fraction $\beta$ of $K_{rs,doy}$ ($d^{-1}$):

$$K_{plant} = \beta\, K_{rs,doy} \qquad (13)$$

We used the measured plant hydraulic conductance from sap flow, leaf water potential, soil water potential, and root observation (Section 2.2.1 above) in the lower rainfed plot to calibrate $\beta$ which was then applied for all plots (Appendix C). $K_{plant}$ and $K_{rs}$ in anisohydric wheat are influenced by soil water availability and crop development. We followed the approach

of Cai et al., (2017) to estimate the root hydraulic conductance ($K_{rs,doy}$) and compensatory root water uptake ($K_{comp}$) based on the total length of the root system below a unit surface area, $TRLD_{doy}$ (m m$^{-2}$), at a given day of year (DOY) (Eq. 14), which is the output from SLIMROOT:

$$TRLD_{doy} = \sum_{i}^{N} RLD_{i,doy}\, \Delta z_i \qquad (14)$$

Assuming the same conductance for all root segments, the root system conductance scales with the TRLD:

$$K_{rs,\, doy} = K_{rs,\, normalized} TRLD_{doy} \qquad (15)$$

where $K_{rs,\, normalized}$ ($d^{-1}$ cm$^{-1}$ cm²) is the root system conductance per unit root length per surface area. For $K_{rs,\, normalized}$, we took

the average value that was obtained by Cai et al., (2018) for the stony soil (F1) and silty soil (F2) sites: 0.2544 10$^{-5}$ ($d^{-1}$ cm$^{-1}$ cm²) (Appendix C).

Many studies included hydraulic conductance along the soil-plant-atmosphere pathway to simulate water transport (Verhoef and Egea, 2014; Wang et al., 2007; Tuzet et al., 2003; Olioso et al., 1996). However, root and plant hydraulic conductance in these studies were assumed constant. In our work, the plant hydraulic conductance varied following the shoot and root

development in the growing season.



**2.3.5 Coupling of water balance and root water uptake models with the crop model**

We carried out a comprehensive comparison of the following modelling approaches for simulating $CO_2$ and $H_2O$ fluxes and crop growth:

- HILLFLOW 1D - Couvreur's RWU - SLIMROOT - LintulCC2 (Co) ;
- HILLFLOW 1D - Feddes' RWU - SLIMROOT - LintulCC2 (Fe)

The photosynthesis and stomatal conductance subroutines, RWU and HILLFLOW 1D water balance model, and evaporative demand (ETP) were run or specified with hourly time steps, while phenology, leaf growth, root growth, and biomass partitioning were updated daily. Before comparing these modelling approaches, we calibrated the original LINTULCC2 model to make sure the model properly described the phenology, LAI, and biomass using the data from the rainfed plot in the silty

soil (F2P2). The same crop parameters and soil parameters were applied for both model configurations (Appendix C, D). All presented flux data (soil water flux, gross assimilation rate, sap flow, leaf water potential) and the simulated outputs were converted from local time to coordinated universal time (UTC) to avoid the confusion in interpretation.

**2.4 Criteria for model comparison and evaluation**

We analysed the performance of two modelling approaches following the approach from (Willmott, 1981): (i) correlation

coefficient (r) (Eq. 16); (ii) the degree to which simulated values approached the observations or index of agreement (I) defined in Eq. (17). This value varies from 1 (for perfect agreement) to 0 (for no agreement); (iii) the root mean square errors (RMSE) was computed to measure the differences between simulated value and observed data (Eq. 18);

$$r = \frac{\sum_{i=1}^{n}\left(Sim_i - \overline{Sim}\right)\left(Obs_i - \overline{Obs}\right)}{\sqrt{\left[\sum_{i=1}^{n}\left(Sim_i - \overline{Sim}\right)^2\right]\left[\sum_{i=1}^{n}\left(Obs_i - \overline{Obs}\right)^2\right]}} \qquad (16)$$

$$I = 1 - \left[\frac{\sum_{i=1}^{n}(Sim_i - Obs_i)^2}{\sum_{i=1}^{n}\left(\left|Sim_i - \overline{Obs}\right| + \left|Obs_i - \overline{Obs}\right|\right)^2}\right] \qquad (17)$$

$$RMSE = \sqrt{\frac{\sum_{i=1}^{n}(Sim_i - Obs_i)^2}{n}} \qquad (18)$$

where Sim and Obs are simulated and measured variables; i is the index of a given variable; $\overline{Obs}$ and $\overline{Sim}$ is the mean of the simulated and measured data; and n is the number of observations;





**2.5 Sensitivity analysis**

The parameters of the SLIMROOT root growth model and the Couvreur RWU model were derived from literature data. However, these parameters may be uncertain and vary between different wheat varieties. In order to evaluate the effect of these parameters on the simulated crop growth and root water uptake, we carried out a sensitivity analysis.

In a first set of simulations, the root length normalized root system conductivity $K_{rs, normalized}$ was varied from 0.1 to 40 times the $K_{rs, normalized} = 0.2554 \ 10^{-5} \ (d^{-1}/cm \ cm^{-2})$ that was estimated by Cai et al., (2018). The root system hydraulic conductance is related to the total root length which depends on the specific weight of lateral and seminal roots. These two parameters are rarely reported, especially for field grown wheat (Noordwijk and Brouwer, 1991). The range of observed specific weight of lateral root in wheat was reported in the range of 0.00406 to 0.00613 g m$^{-1}$ (Noordwijk and Brouwer, 1991). Huang et al., (1991) found that the specific weight of seminals root of winter wheat grown under controlled soil chamber conditions decreased from 0.023 to 0.0052 g m$^{-1}$ when air temperature increased from 10 to 30°C. The values of 0.015 and 0.0035 g m$^{-1}$ are often used for specific weights of seminal and lateral roots, respectively in crop growth simulations of wheat cultivars (Mboh et al., 2019; Jamieson and Ewert, 1999). In a second set of simulations, the specific weight of lateral roots was subjected to change from 0.002, 0.003, 0.0035, 0.004, 0.005, 0.006, and 0.007 g m$^{-1}$ while specific weight of seminal roots was the same (0.015 g m$^{-1}$) for all simulations. For the third set of simulations, specific weight of lateral root was kept at 0.0035 g m$^{-1}$ while the specific weights of seminal root varied from 0.005, 0.0075, 0.01, 0.0125, 0.015, 0.0175, 0.02, and 0.0225. In the last sensitivity exercise, the critical leaf water potential $\Psi_{thresholds}$ was varied between -120 m and -260 m.

**3 Results and discussion**

In the first section, we discuss the performance of the two coupled root-shoot models with Couvreur RWU model (Co model) and Feddes RWU model (Fe model). The comparative analysis firstly focuses on simulating crop growth and root development under different water conditions and soil types. Next, the simulated transpiration reduction, soil water dynamics, RWU, and gross assimilation rate are presented and discussed. The explicitly simulated $K_{plant}$ by the Co model in the different soils and treatments is compared with direct estimates of $K_{plant}$ from measurements. In the second part, we discuss the sensitivity analysis of the Co model to understand the effects of changing $K_{rs, normalized}$, specific weight of seminal and lateral root, and $\Psi_{threshold}$ on the simulated biomass growth and RWU under different soils and water regimes.

**3.1 Comparison of Couvreur and Feddes's RWU model**

**3.1.1 Root and shoot (biomass and LAI) growth**

Fig. 2 shows the dry matter and LAI simulated by the Co and Fe model versus the measured data. The difference between the two samples of the two different rows for each sampling day indicated the heterogeneity in crop growth even within a small treatment plot. Biomass and LAI simulated by the Co and Fe models were in close agreement with observations. The $r^2$ of Co and Fe models were 0.91 and 0.86, respectively, for biomass while 0.76 and 0.75, respectively, for LAI (Table 1). However,





both models overestimated dry matter and LAI production in the irrigated and rainfed stony plots whereas biomass and LAI were underestimated in the sheltered silty plot. This suggests that water stress in the sheltered silty plot was overestimated. For the irrigated stony soil plot, in which the water content stayed high due to the frequent rainfall events and the additional irrigation, it is unlikely that the lower growth is due to water stress. The later start of the growth after the winter could be due

to the effects of soil strength and lower soil temperature on crop development in the stony field that were not captured by the model. Soil hardness could constrain root growth while the higher stone content possibly resulted in slower warming up of the soil in spring than the silty soil which in turn slowed down root and crop development.

[Insert Fig. 2 here]

For the stony plots, the Fe and Co models gave similar results whereas for the silty soil, the Co model reproduced the biomass

and LAI better than the Fe model. Although the statistical parameters (r² and RMSE) for the silty soil plots show only a slightly better fit of the Co than of the Fe model, there is a remarkable qualitative difference between the models. The Fe model simulated lower biomass and leaf area in the silty soil than in the stony soil, which is opposite to the observations. The Co model simulated similar biomass and LAI in the irrigated and rainfed plots of the silty and stony soils and higher biomass and LAI in the sheltered plot in silty soil than in the stony soil, which is in closer agreement with the observed differences in

biomass and LAI between the two soils. The simulated effect of the soil type on the crop growth was qualitatively correct for the Co model but incorrect for the Fe model.

[Insert Table 1 here]

Fig. 3 displays the observed root length densities from minirhizotube observations and the simulated ones. Higher root length densities were observed and simulated in the silty soil than in the stony soil. The model simulated smaller root densities in the

stony soil because a larger specific weight of the roots was considered for the stony than for the silty soil. The simulated root density profiles showed the highest root densities near the surface whereas the observed profiles, especially in the silty soil, showed higher densities in the deeper soil layers. The model simulated smaller root length densities in the sheltered than in the other plots of both the stony and silty soils. This is a consequence of the lower biomass growth that was simulated in the sheltered plots. For the stony soil, this corresponds with the observations that also showed lower root length densities in the

sheltered than in the other plots. However, for the silty plot, the opposite was observed. For both the simulations and the observations, we compared the ratio of total root lengths in a certain plot and treatment to the total root length in rainfed stony plot F1P2 (Appendix F). In the stony plots the ratios of the observed total root length to the reference were close to 1 but the simulated total root length in the sheltered plot was smaller than one. The ratios of the total root lengths in the silty plot to the reference were for all plots larger than one. Nevertheless, the ratios of observed root lengths were larger (2.27 - 4.03) than

those of the simulated ones (1.04 - 1.67). The observed ratios were larger for the sheltered plot than for the other plots in the silty soil whereas the opposite was simulated by the models. Predefined ratios of root and shoot biomass allocation for a given growth period and a source driven root growth (Goudriaan and van Laar, 1994) in our models do not allow a shift in carbon allocation to root (for more root growth) in response to water stress. However, this should be not too emphasized because the



observed imaged root data from rhizotubes for driving the root length might have potential errors and uncertainties (Cai et al.,
360    2018).

[Insert Fig. 3 here]

### 3.1.2 Transpiration reduction, soil water dynamic, RWU, and gross assimilation rate

Fig. 4a and 4b show the simulated reduction of the transpiration compared to the potential transpiration, $f_{wat}$, by the Fe and Co
models (mid of March until harvest) and Fig. 4c and 4d show the simulated potential and the simulated and measured actual
transpiration rates from the end of April until harvest. The Fe model simulated more water stress than the Co model and a more
pronounced and earlier stress in the silty than in the stony soil. As a consequence, the simulated transpiration rates by the Fe
model were generally lower than the simulated ones by the Co model. According to the $f_{wat}$ factors, also the Couvreur model
simulated more water stress in the silty soil than in the stony soil. The effect of $f_{wat}$ on the cumulative transpiration and growth
depends also on the timing of the lower $f_{wat}$ values. At the beginning of the growing season when the LAI and potential
transpiration are low, the impact of a lower $f_{wat}$ on the cumulative transpiration and growth is lower than later in the growing
season. These results are in contrast with findings by Cai et al., (2017) and Cai et al., (2018) who found that there was no water
stress simulated in the silty soil in 2014 by the Co and Fe models. However, the studies from Cai et al., (2018) used the
measured root distributions instead of the simulated ones from the root-shoot model. Therefore, in their simulations, the crop
had more access to water in the deeper soil layers. Second, they used the Feddes-Jarvis model, which accounts for root water
uptake compensation. This could explain why they did not simulate water stress in the silty plot with the Feddes model. Thirdly,
weather conditions and irrigation applications were different in their study in 2014 (less drier) from our experimental season
in 2016.

[Insert Fig. 4 here]

According to Fig. 4c and 4d, during the time when sap flow could be measured (end of May until harvest), the stress factors
did not differ a lot between the Fe and Co models. For the rainfed and irrigated plots in the silty soil, the Fe model predicted a
stronger reduction in transpiration near the end of the growing season than the Co model. This resulted in a smaller cumulative
transpiration predicted by the Fe than by the Co model over the measurement period in these treatments (Fig. 5). Although this
gives the impression that the Co model is better in agreement with the measurements in these treatments, Fig. 4d indicates that
this is due to compensating errors. Both models underestimate the measured sap flow in the beginning of the measurement
period and overestimate it towards the end, and the Co model overestimates more than the Fe model. This overestimation is
due to an overestimation of the LAI by both models near the end of the growing season (Fig. 2b). The reduction of the
transpiration in the sheltered plots of the two soils compared to the other treatments is predicted relatively well but the Fe
model predicted more stress and a stronger reduction in transpiration than the Co model, especially in the silty soil. For this
treatment, the Co model, which simulated less stress (larger $f_{wat}$ factors), predicted the cumulative transpiration and how it
differed between the two soil types better than the Fe model.





[Insert Fig. 5 here]

Simulated transpiration in all treatments and both soils are plotted versus the sap flow measurements in Fig. 6. On average, the two models slightly underestimated measured $T_{act}$ (Fig. 4c and 4d). This was also found in the study by Cai et al., (2018) where sap flow was measured in winter wheat in 2014. However, in their study, there was a rather constant offset between the

simulations and the sap flow data. One reason could be that in our study we used the simulated LAI values whereas Cai et al., (2018) used the measured LAI values. In the stony plots, the measured LAIs are overestimated by the simulations so that one would expect an overestimation of the transpiration by the model. The opposite holds true for the silty plot. The overestimation of the LAI at the end of growing season resulted in an overestimation of the transpiration in non-sheltered plots in both soil types. Because of the small size and hollow stem of wheat plants (Langensiepen et al., 2014), it is difficult to install the micro-

sensors and measure the temperature variation for the thin wheat stem with high time frequency under ambient field conditions. In addition, the sap flow in a single tiller is also influenced by spatial variation in environmental conditions. The variability of stem development also results in a significant stem-to-stem variability in sap flow (Cai et al., 2018). The $r^2$ of simulated RWU from the Co and Fe models versus sap flow are 0.62 and 0.66, respectively (Table 1 and Fig. 6a) indicating that our coupled models have adequate performance in RWU simulation. Measuring gas exchange with closed chamber concentration

measurements can significantly alter the microclimatic conditions within the chamber, especially at times of high exchange rate. However using regression functions at the starting point of measurement intervals reduces absolute errors (Langensiepen et al., 2012). The simulated Pg from two models matched relatively well with the gross assimilation rate measured by a manually closed-canopy chamber with $r^2$ of 0.63 and 0.61 for Co and Fe, respectively (Table 1 and Fig. 6b).

[Insert Fig. 6 here]

The differences in simulated stress between the different models were more pronounced in May (Fig. 4) when no sap flow data were available. The Co model predicted less stress and more RWU than the Fe model in May, especially in the rainfed and irrigated plots of the silty soil. The larger stress simulated by the Fe model in the rainfed and irrigated silty plots resulted in a smaller increase in biomass that was simulated in May by the Fe than by the Co model (Fig. 2a). The measurements of growth in the silty soil do not suggest that there was water stress in these plots in the silty soil indicating that the Co model

better transpiration and growth for these cases than the Fe model. Another way to test the RWU simulated by the different models is to compare the simulated soil water contents (Fig. 7). The Co and Fe models were able to simulate both dynamic and magnitude of SWC over different soil depths in the different water treatments (average of RMSEs over all soil depths was 0.06 for both models, Appendix G). The Co and Fe models displayed lower water contents than the measured ones in the deeper layers at the late growing season (i.e. depth 80 and 120 cm) (Fig. 7). This could be due to the free drainage bottom

boundary condition in the HILLFLOW water balance model, which implies that the water can only leave the soil profile but no water can flow in it. Capillary rise in the soil can keep the lower layers relatively wet (Vanderborght et al., 2010). In our simulation, the use of a soil depth of 1.5 m may not be deep enough to capture this effect. The simulated SWC were however very similar for both models. The larger RWU simulated by the Co than by the Fe model in the silty soil in May resulted in





slightly lower simulated water contents by the Co model. But, the differences in simulated water contents by the two models

were much smaller than the deviations from the observed water contents.

[Insert Fig. 7 here]

For a few selected days, the diurnal course of $T_{act}$ (or RWU) gross assimilation rate (Pg), and leaf pressure head were measured. The measured and simulated data are shown in Fig. 8. Both Co and Fe models could mimic the daytime fluctuation of RWU and Pg in the sheltered plot the stony soil which is consistent with the adequate simulation of root growth (Fig. 3, F1P1) and

SWC dynamic (Fig. 7a, F1P1). When the simulated $\psi_{leaf}$ reached $\psi_{threshold} = - 200$ m, the simulated RWU and Pg by the Co model showed a plateau (26 May in Fig. 8a, 8c, and 8e). Using the leaf water pressure head threshold as an indication of water stress effects on stomata, Tuzet et al., (2003) and Olioso et al., (1996) also reported a considerable drop of Pg and transpiration. The sharp drop of simulated RWU and Pg which is in contrast with measurement on the same day in the sheltered plot in silty soil illustrated that both models overestimated the water stress. This related to the underestimation of both root growth (Fig.

3, F2P1) and SWC (Fig. 7b, F2P1) in the deeper soil layers by two models.

[Insert Fig. 8 here]

### 3.1.3 Whole plant hydraulic conductance from Couvreur RWU model

The Couvreur RWU model considers the root hydraulic conductance which relies on absolute root length. The root hydraulic conductance is used to upscale to whole plant hydraulic conductance. The simulated $K_{plant}$s reproduced the measured ones in

the different treatments quite well (Fig. 9). Our measured $K_{plant}$ ranged from $1.5 \times 10^{-5}$ to $10.2 \times 10^{-5}$ $d^{-1}$ (Fig. 9). These values are in the same order of magnitude as values reported by Feddes and Raats, (2004) for ryegrass ranging from $6 \times 10^{-5}$ to $20 \times 10^{-5}$ $d^{-1}$. The simulated $K_{plant}$ from our coupled root and shoot Co model followed the root growth and reached a maximum at around anthesis. $K_{plant}$ reduces toward the end of the growing season due to root death. For the sheltered plot of the silty field, we would expect based on the root density measurements (Fig. 3), the highest $K_{plant}$ of all treatments. However, this was not

observed in the measurements. Based on the measured total root lengths, we would also expect that $K_{plant}$ of the sheltered plot in the stony field should be similar to $K_{plant}$ in the other plots of the stony field. But, $K_{plant}$ was clearly lower in the sheltered plot of the stony field than in the other treatments in the stony field. In the model simulations, the lower $K_{plant}$ in the sheltered plots compared to the other plots in the same fields was due to a lower simulated total root length. Since the differences in observed total root lengths were smaller (stony soil) or opposite (silty soil) to the differences in simulated total root lengths,

the smaller observed $K_{plant}$ in the sheltered plots have probably other causes that are not considered in the model. A potential candidate is the resistance to water flow from the soil to the root in the soil, which increases considerably when the soil dries out, as was the case in the sheltered field plots.

[Insert Fig. 9 here]





## 3.2 Effects of changing root hydraulic conductance and leaf water pressure head thresholds

We conducted three sets of simulations. In the first set of simulations $K_{rs, normalized}$ was subjected to change. Fig. 10 illustrates the sensitivity of Co model to $K_{rs, normalized}$ in terms of above-ground biomass at harvest and cumulative RWU (from 15 March to harvest) for the different water treatments and soil types. For the rainfed and irrigated plots, an increase in $K_{rs, normalized}$ does not lead to a substantial increase in RWU and above ground biomass. This is a trivial consequence of the fact that water is not (irrigated plots) or only slightly (rainfed plots) limited in these cases. For the stony soil, a decrease of $K_{rs, normalized}$ by a certain factor leads to a stronger decrease in RWU and biomass than in the silty soil. This indicates that in the stony soil, less water is 'accessible' so that a decrease in root water uptake capacity by the crop has a stronger impact on RWU and biomass production than in the silty soil. For the sheltered plots, RWU and biomass production increases with $K_{rs, normalized}$ suggesting that increasing the water uptake capacity by the plants would increase the uptake and growth. But, increasing $K_{rs, normalized}$ by the same factor had a smaller relative effect on the RWU and biomass production than decreasing $K_{rs, normalized}$.

[Insert Fig. 10 here]

Decreasing the specific weight of lateral and seminal roots increases the specific root length and thus total root length of root system, total root system hydraulic conductance, and thus and whole plant hydraulic conductance. However, for the considered range of specific weights, there was only a minor increase of above dry biomass and RWU (Fig. 10c-f). Reducing the specific root length by increasing the specific weights of lateral and seminal roots caused a stronger reduction in biomass and RWU, especially for the seminal root in the stony soil. High values of $\Psi_{threshold}$ led to more water stress and a sharp decrease in stomatal conductance and photosynthesis when $\Psi_{leaf}$ was limited to its thresholds (Fig. 10g & h). Our results suggested that $\Psi_{threshold}$ at -120 m or -140 m could overestimate the water stress while the $\Psi_{threshold}$ at -260 m could underestimate the stress.

The impact of the change of the root segment conductance, specific weight of roots, and the leaf pressure head threshold at which stomata close on RWU and above ground biomass is amplified by the positive feedback between the above ground biomass, the root biomass, the total root length, the root system hydraulic conductance, and finally $K_{plant}$. Considering these interactions and feedbacks is important to evaluate the impact of changing a certain property of the crop on its performance in different soils and under different conditions.

The root: shoot ratios of modern cultivars were lower than old wheat cultivars (Zhao et al., 2005; Siddique et al., 1990). However, the hydraulic conductance of single roots and the whole root system were increased in the modern cultivars and inversely correlated to the root: shoot ratio (Zhao et al., 2005). This indicates the water uptake ability of wheat roots was improved from wild to modern varieties during evolution with larger root system hydraulic conductance. In addition, recently, contrasting stomatal regulations were reported for different winter wheat genotypes that are related to the genotype-specific synthesis of ABA (Gallé et al., 2013). Plants with a high stomatal sensitivity to leaf water potential (pressure head) are then modelled with a higher reference (or critical) leaf water potential (-1.2 MPa) (or -120 m) while for species like wheat or lupine which are more tolerant to water stress a lower reference leaf water potential was used (i.e. -1.9 or -2.6 MPa) (equivalent to -190 m or -260 m, respectively) (Tuzet et al., 2003).





The impact of changing root system properties or stomatal sensitivity to water potential on root water uptake, stress, and crop growth cannot be assessed by a model that is not sensitive to these crop properties. Different to the Co model the Fe model is

not sensitive to the total root length, the normalized root conductance, the specific root weight, and the leaf water potentials at which stomata close. Therefore, the impact of introducing crop varieties with new properties cannot be assessed by this type of model. Only with the Co model the impact of the crop properties on growth and drought resilience can be studied.

## 4 Conclusion

We evaluated two different root water uptake modules of a coupled soil water balance and crop growth model. One root water

uptake model was the often used Feddes model whereas the other, the Couvreur RWU model represents a "mechanistic" RWU model that explicitly simulates the continuum in water potential from soil to root, and to leaf based on the whole plant hydraulic conductance. The whole plant hydraulic conductance was calculated from the total root length and a root segment hydraulic conductance. All parameters of the model were derived from literature and from a previous study that was carried out at the same experimental site but for another growing season (Cai et al., 2018). Only one parameter of the Co model, i.e. a factor that

was used to scale the root system conductance to the whole plant hydraulic conductance was manually adjusted. The soil, crop, and RWU parameters were applied to simulate crop biomass, LAI, root densities, and depth distributions, soil moisture contents, leaf water potentials, transpiration, and assimilation rates in two different soils and with each three different water treatments.

Overall, the measured biomass growth, LAI development, soil water contents, leaf water potentials, and transpiration rates

were well reproduced by both models. But, the Fe model incorrectly predicted more water stress and less growth in the silty soil than in the stony soil whereas the opposite was observed. The Co model was able to predict the response of the crop to the different water stress conditions in the different soils and treatments. This was explained by more root growth in the silty soil which increased the root/plant hydraulic conductance, as was confirmed by direct estimates of the plant system conductances, and reduced the water stress. A mechanistic model that is based on plant hydraulics and links root system properties to RWU,

water stress, and crop development can evaluate the impact of certain crop properties on crop performance in different environments and soils. The Fe model does not account for the higher plant conductance in the silty soil where more roots were simulated than in the stony soil. In addition, the Fe model does not consider root water uptake compensation which reduces water stress. In other words, the Feddes approach did not possess the flexibility as compared to Couvreur model in simulating RWU for different soil and water conditions.

Given the important role of root system properties for RWU and plant water stress, modelling root development and how it responds to water deficiency is crucial to predict the impact of water stress on crop growth. In this study, a higher total root length was simulated in the silty soil than in the stony soil because a higher specific root length was found for root growth in the silty soil. This can be considered as an extra relationship that requires attention in crop modelling. Crop growth models will need to consider soil specific calibration to account for differences in specific root length with soil. Alternatively, a more

mechanistic description of root growth that predicts root specific length would reduce the amount of calibration in crop growth





models. Another aspect in demand of improvement is the prediction of the root distributions with depth. In our simulations, highest root densities were simulated in the top soil whereas the observations showed higher densities in the deeper soil layers. However, these observations were obtained from minirhizotubes and more validation with direct measurements of root distributions would be required. Finally, the model did not consider changes in carbon allocation to the root system that are

triggered by stress. Therefore, the model simulated less roots in the water stressed sheltered plot of the silty soil whereas more roots were observed in this plot compared with the other plots in this soil. A more mechanistic description of the carbon allocation as a function of soil water conditions would be needed to refine the prediction of responses of root development to water stress.

Future research should focus on testing the newly coupled model (Couvreur - HILLFLOW - LINTULCC2) for other wheat

genotypes and crop types (isohydric like maize) and for a wider range of soil and climate conditions. Further improvements should particularly be targeted leaf area simulation. Improving the modelling of leaf growth should result in better simulations of LAI and more accurate estimates of energy fluxes at canopy level.

**Appendices**

**Appendix A: Leaf photosynthesis and stomatal conductance calculation**

$$AMAX_{l,t} = \frac{VCMAX_{l,t}(Ci_{l,t} - \Gamma^*)}{Ci_{l,t} + KMC\left(1 + \frac{O_2}{KMO}\right)} fwat \qquad (A1)$$

$$EFF_{l,t} = \frac{J}{2.1} \frac{Ci_{l,t} - \Gamma^*}{4.5(Ci_{l,t} + 2\Gamma^*)} \qquad (A2)$$

$$FGR_{l,t} = AMAX_{l,t}\left(1 - e^{-I_{l,t}\frac{EFF_{l,t}}{AMAX_{l,t}}}\right) \qquad (A3)$$

$$Ci_{l,t} = Ca - \left(FGR_{l,t}\frac{1}{gs_{l,t}}\right) \qquad (A4)$$

$$gs_{l,t} = a_1 + \frac{b_1 FGR_{l,t}}{(Ci_{l,t} - \Gamma^*)(1 + \frac{DS_{l,t}}{D_0})} fwat \qquad (A5)$$

AMAX is light saturated leaf photosynthesis ($\mu M\ CO_2\ m^{-2}\ s^{-1}$); VCMAX is maximum carboxylation rate of Rubisco enzyme ($\mu M\ m^{-2}\ s^{-1}$); Ci is intercellular $CO_2$ concentration ($\mu M\ mol^{-1}$); Ca is atmospheric $CO_2$ concentration ($\mu M\ mol^{-1}$); KMC is Michaelis-Menten constant for $CO_2$ ($\mu M\ mol^{-1}$); KMO is Michaelis-Menten constant for $O_2$ ($\mu M\ mol^{-1}$); $O_2$ is atmospheric oxygen concentration ($\mu M\ mol^{-1}$); $\Gamma^*$ is $CO_2$ compensation point ($\mu M\ mol^{-1}$); EFF is quantum yield ($\mu M\ CO_2\ MJ^{-1}$); J is





conversion energy from radiation to mole photon (mole photons MJ$^{-1}$); FGR is leaf photosynthesis rate ($\mu$M CO$_2$ m$^{-2}$ s$^{-1}$); I is

the total absorbed flux of radiation (MJ m$^{-2}$ s$^{-1}$); gs is bulk stomatal conductance (mol m$^{-2}$ s$^{-1}$); a$_1$ is residual stomatal
conductance (mol m$^{-2}$ s$^{-1}$) when FGR = 0; b$_1$ is fitting parameter (-); DS is the vapor pressure deficit at the leaf surface (Pa);
D$_0$ is empirical coefficient reflecting the sensitivity of the stomata to VPD (Pa); l is sub-indices indicates canopy layer (sunlit
and shaded leaf) (-); t is sub-indices indicates time of the day (-); fwat is water stress factor for stomatal conductance and
maximum carboxylation rate (-);


**Appendix B: Scale up leaf stomatal conductance to canopy resistance in hourly simulation**

To scale up from leaf stomatal conductance to canopy and for computation efficiency, we approximate the integrals

$$\int_0^{LAI} f(l)dl$$

By Gaussian quadrature $LAI \sum_{j=1}^{5} w_j f(LAI\ x_j)$ where x$_j$ are the nodes and w$_j$ the weights of the 5-point gaussian quadrature.

LAI is the leaf area index and f is a function dependent on leaf area for instance gsH2O.

The above mentioned bulk stomatal conductance to CO$_2$ (gs$_{l,t}$ - mol m$^{-2}$ s$^{-1}$) of sunlit and shaded leaf to stomatal conductance
was converted to stomatal conductance to H$_2$O (m s$^{-1}$) based on the molar density of air.

$$gsH2O_{sun} = 1.56 * gs_{sun}/41.66 \qquad (B1)$$

$$gsH2O_{shade} = 1.56 * gs_{shade}/41.66 \qquad (B2)$$

Leaf stomatal conductance to H$_2$O (m s$^{-1}$) was calculated based on fraction of sunlit leaf area FSLLA

$$gsH2O_{leaf} = gsH2O_{sun} * FSLLA + gsH2O_{shade}(1 - FSLLA) \qquad (B3)$$

Because the hourly weather input was used in hourly simulations (Feddes and Couveur), thus there was no Gaussian integration
over time degree. The hourly canopy conductance HourlyGSCropH2O (m s$^{-1}$) was calculated in Eq. (B4)

$$HourlyGSCropH2O = LAI * \sum_{j=1}^{5} w_j\ gsH2O_{leaf} \qquad (B4)$$

Hourly canopy resistance (s m$^{-1}$) was the reciprocal of hourly canopy conductance

$$Hr_s = 1/HourlyGSCropH2O \qquad (B5)$$

Assuming the leaf cuticle resistance and soil surface resistance were minor and neglected, the calculated canopy resistance
(Hr$_s$) with fwat = 1 was directly used to calculate hourly crop evapotranspiration (ETP) hourly using Penman-Monteith (Eq.
B6).



$$ETP = \frac{\Delta(R_n - G) + \rho_a c_p \frac{(e_s - e_a)}{r_a}}{\lambda\left(\Delta + \gamma\left(1 + \frac{Hr_s}{r_a}\right)\right)}$$

(B6)

$R_n$ is net radiation (MJ m$^{-2}$ h$^{-1}$) ; G is soil heat flux (MJ m$^{-2}$ h$^{-1}$); $e_s$ is saturation vapor pressure at the air temperature (kPa); $e_a$ is actual vapor pressure at the air temperature (kPa); ρa is mean air density at constant pressure( kg m$^{-3}$); $c_p$ is the specific heat at constant pressure of the air (1.013 10$^{-3}$ MJ kg$^{-1}$ ∘C$^{-1}$); $\Delta$ is slope of the saturation vapor pressure-temperature relationship (kPa ∘C$^{-1}$); γ is the psychrometric constant of instrument (kPa ∘C$^{-1}$), Hr$_s$ is surface resistance (s m$^{-1}$); $r_a$ is the aerodynamic

resistance (s m$^{-1}$); λ is the latent heat of vaporization (2.45 MJ kg$^{-1}$).

## Appendix C: Crop parameters used in the modelling work

| Sub-models | Parameters | Explanation (unit) | Stony | Silty | Reference |
|---|---|---|---|---|---|
| LINTULCC2 | VCMAX | Maximum carboxylation rate of Rubisco at 25°C (μM m$^{-2}$ s$^{-1}$) | 62.1 | | Yin et al., (2009) |
| | Ca | Atmospheric $CO_2$ concentration (μM mol$^{-1}$) | 410 | | |
| | RGRL | Relative growth rate of leaf area during exponential growth (°Cd)$^{-1}$ | 0.007 | | van Laar et al., (1997) |
| | LAICR | Critical leaf area index (-) | 5 | | van Laar et al., (1997) |
| SLIMROOT | RSROOT$_{max}$ | Maximal elongation rate of seminal roots per day (m d$^{-1}$) | 0.03 | | Watt et al., (2006) |
| | DRRATE | Daily fraction of dying roots (-) | 0.008 | | |
| | RINPOP | Number of emerged plants per square meter (number m$^{-2}$) | 350 | | |
| | MAXDEP | Maximum root depth (m) | 1.5 | | |
| | NRSPP | Number of seminal root per plant (number plant$^{-1}$) | 3 | | Shorinola et al., (2019); Huang et al., (1991) |
| | WLROOT | Specific weight for lateral root (g m$^{-1}$) | 0.0061 | 0.004 | Jamieson and Ewert, (1999); Noordwijk and Brouwer (1991) |
| | WSROOT | Specific weight of seminal root (g m$^{-1}$) | 0.02 | 0.015 | Jamieson and Ewert, (1999); Huang et al., (1991) |
| Feddes | hlim1 | Soil water potential at anaerobic point (m) | 0 | | Cai et al., (2018) |
| | hlim2 | Soil water potential where optimum condition for transpiration (m) | -0.01 | | Cai et al., (2018) |
| | hlim3h | Soil water potential for higher transpiration rate (m) | -2.79 | | Cai et al., (2018) |
| | hlim3l | Soil water potential for lower transpiration rate (m) | -7.47 | | Cai et al., (2018) |
| | hlim4 | Soil water potential at wilting point (m) | -160 | | Cai et al., (2018) |
| | T$_{pot3h}$ | Higher transpiration rate (m d$^{-1}$) | 0.0048 | | Cai et al., (2018) |
| | T$_{pot3l}$ | Low transpiration rate (m d$^{-1}$) | 0.00096 | | Cai et al., (2018) |



| Couvreur | $\Psi_{threshold}$ | Threshold of leaf water potential for specific plant (m) | -200 | Cochard, (2002); Tardieu and Simonneau, (1998) |
| | $K_{rs, normalized}$ | Initial normalized root hydraulic conductance (d$^{-1}$/cm cm$^{-2}$) | 0.2544 10$^{-5}$ | Cai et al., (2018) |
| | $K_{comp, normalized}$ | Initial normalized compensatory hydraulic conductance (d$^{-1}$/cm cm$^{-2}$) | 0.0636 10$^{-5}$ | Cai et al., (2018) |
| | $\beta$ | Fraction to upscale from $K_{rs}$ to $K_{plant}$ (-) | 0.55 | |

**Appendix D: Soil physical parameters at the top (0-30 cm) and subsoil (30-150 cm)**

| Soil types | Layers | α | n | l | θ_r | θ_s | ks |
| | | (m$^{-1}$) | (-) | (-) | (m$^3$ m$^{-3}$) | (m$^3$ m$^{-3}$) | (m s$^{-1}$) |
| Stony | Top soil | 3.61 | 1.386 | 3.459 | 0.0430 | 0.3256 | 10.7*10$^{-6}$ |
| | Sub soil | 4.95 | 1.534 | 3.459 | 0.0543 | 0.2286 | 5.83*10$^{-8}$ |
| Silty | Top soil | 2.31 | 1.292 | 1.379 | 0.1392 | 0.4089 | 1.16*10$^{-6}$ |
| | Sub soil | 0.50 | 1.192 | 1.379 | 0.1304 | 0.4119 | 1.73*10$^{-6}$ |

The θ_r and θ_s are residual and saturation soil water content, respectively; α, n, l are empirical coefficients affecting the shape of the van Genuchten hydraulic functions ; ks is saturated hydraulic conductivity of the soil

**Appendix E: Feddes root water uptake model**

The root water uptake in HILLFLOW 1D model which is limited by soil water content in the root zone calculated by reduction of potential transpiration ($T_{pot}$). The semi-empirical reduction function $\alpha(\Psi_{m,i})$ is derived from matrix potential head (Feddes

et al., 1978). The $\alpha(\psi_{m,i})$ also depends on $T_{pot}$ because $\psi_3$ (soil pressure head where optimum condition for transpiration) is calculated via piecewise linear function of $T_{pot}$ (Wesseling and Brandyk, 1985). The root water uptake was calculated based on relative root length density which is output from the SLIMROOT root growth model.

$$\alpha\left(\psi_{m,i}\right) = \begin{cases} 0 & \psi_{m,i} \geq \psi_1, \psi_{m,i} \leq \psi_4 \\ (\psi_{m,i} - \psi_1)/(\psi_2 - \psi_1) & \psi_2 \leq \psi_{m,i} \leq \psi_1 \\ 1 & \psi_3 \leq \psi_{m,i} \leq \psi_2 \\ (\psi_{m,i} - \psi_4)/\psi_3 - \psi_4) & \psi_4 \leq \psi_{m,i} \leq \psi_3 \end{cases}$$

(F1)

$$\psi_3 = \begin{cases} \psi_{3h} & T_{pot} > T_{3h} \\ \psi_{3h} + \dfrac{(\psi_{3l} - \psi_{3h})(T_{3h} - T_{pot})}{(T_{3h} - T_{3l})} & T_{3l} < T_{pot} < T_{3h} \\ \psi_{3l} & T_{pot} < T_{3l} \end{cases}$$

(F2)





α($\Psi_{m,i}$) transpiration reduction as function of matrix potential head (-); $\Psi_1$ is soil water potential at anaerobic point (m); $\Psi_4$ is soil water potential at wilting point (m); $\Psi_2$ and $\Psi_3$ is soil water potential where optimum condition for transpiration (m); $T_{pot}$

is potential transpiration (m d$^{-1}$); $\Psi_{3h}$ is soil water potential for higher potential transpiration rate (m); $T_{3h}$ is higher potential transpiration rate (m d$^{-1}$); $\Psi_{3l}$ is soil water potential for lower potential transpiration rate (m); $T_{3l}$ is lower potential transpiration rate (m d$^{-1}$).

**Appendix F:**

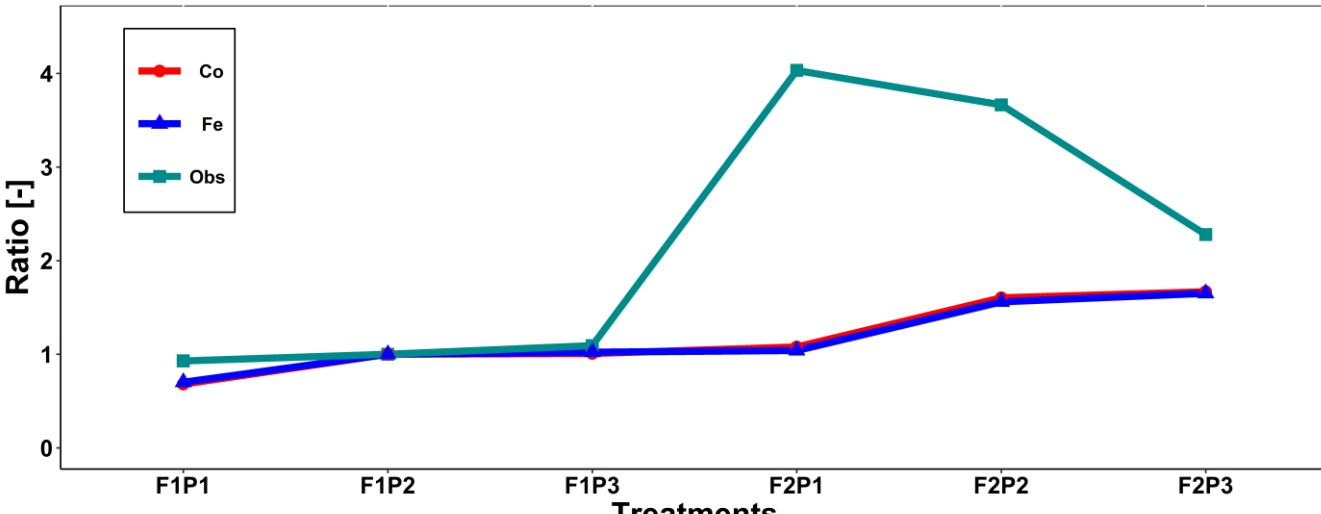


Appendix F: Comparison ratio of the observed total root length from rhizotubes to observed total root length from F1P2 (green line with squares) and ratio of simulated total root length to the simulated total root length from F1P2 on 11 July 2016 (DOY 193) from Couvreur (Co, solid red, dots), and Feddes (Fe, solid blue, triangles) model at the sheltered (P1), rainfed (P2), and irrigated (P3) plots of the stony soil (F1) and the silty soil (F2)





**Appendix G:** Statistic RMSEs of soil water content simulated by the two models: the Couvreur (Co) and Feddes (Fe) in the sheltered (P1), rainfed (P2), and irrigated (P3) plots of the stony soil (F1), and the silty soil (F2). RMSE is $cm^3\,cm^{-3}$

|  | | F1 | | F2 | |
| --- | --- | --- | --- | --- | --- |
|  | Depth (cm) | Co | Fe | Co | Fe |
| P1 | 10 | 0.09 | 0.09 | 0.08 | 0.08 |
|  | 20 | 0.08 | 0.08 | 0.06 | 0.05 |
|  | 40 | 0.04 | 0.04 | 0.07 | 0.07 |
|  | 60 | 0.07 | 0.07 | 0.03 | 0.03 |
|  | 80 | 0.08 | 0.08 | 0.03 | 0.03 |
|  | 120 | 0.03 | 0.03 | 0.06 | 0.05 |
| P2 | 10 | 0.10 | 0.10 | 0.09 | 0.08 |
|  | 20 | 0.10 | 0.10 | 0.07 | 0.07 |
|  | 40 | 0.06 | 0.06 | 0.07 | 0.06 |
|  | 60 | 0.06 | 0.06 | 0.05 | 0.05 |
|  | 80 | 0.05 | 0.04 | 0.06 | 0.06 |
|  | 120 | 0.06 | 0.06 | 0.06 | 0.05 |
| P3 | 10 | 0.11 | 0.12 | 0.10 | 0.11 |
|  | 20 | 0.12 | 0.12 | 0.08 | 0.08 |
|  | 40 | 0.08 | 0.08 | 0.09 | 0.08 |
|  | 60 | 0.07 | 0.07 | 0.06 | 0.05 |
|  | 80 | 0.05 | 0.06 | 0.06 | 0.06 |
|  | 120 | 0.03 | 0.03 | 0.07 | 0.07 |

*Data availability.* The meteorological data were collected from a weather station in Selhausen (Germany) which belongs to
TERENO network of terrestrial observatories. Weather data are freely available in TERENO data portal (http:
www.tereno.net). The data which were obtained from the rhizotron facilities (under and above ground) are available from the
corresponding author on reasonable request and with permission from TR32 database (www.tr32db.uni-koeln.de).

*Competing interests.* The authors declare that they have no conflict of interest

*Acknowledgements.* This research was financed by the German Science Foundation (DFG) within framework of Transregional
Collaborative Research Center 32" Patterns in Soil-Vegetation-Atmosphere-Systems" (TR32, www.tr32.de). We thank
Gunther Krauss for the technical support in modelling configurations. We thank our student assistants for their enthusiastic
help for data collection in the field. We also thank Andrea Schnepf, Gaochao Cai, Miriam Zoerner, and Shehan Tharaka
Morandage for providing soil water content, soil water potential, and root growth data.



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

Figure 1: Description of the location of field experiment and set up of water treatments in the stony soil (F1) and silty soil (F2). P1, P2, and
P3 are the sheltered, rainfed, and irrigated plots, respectively.



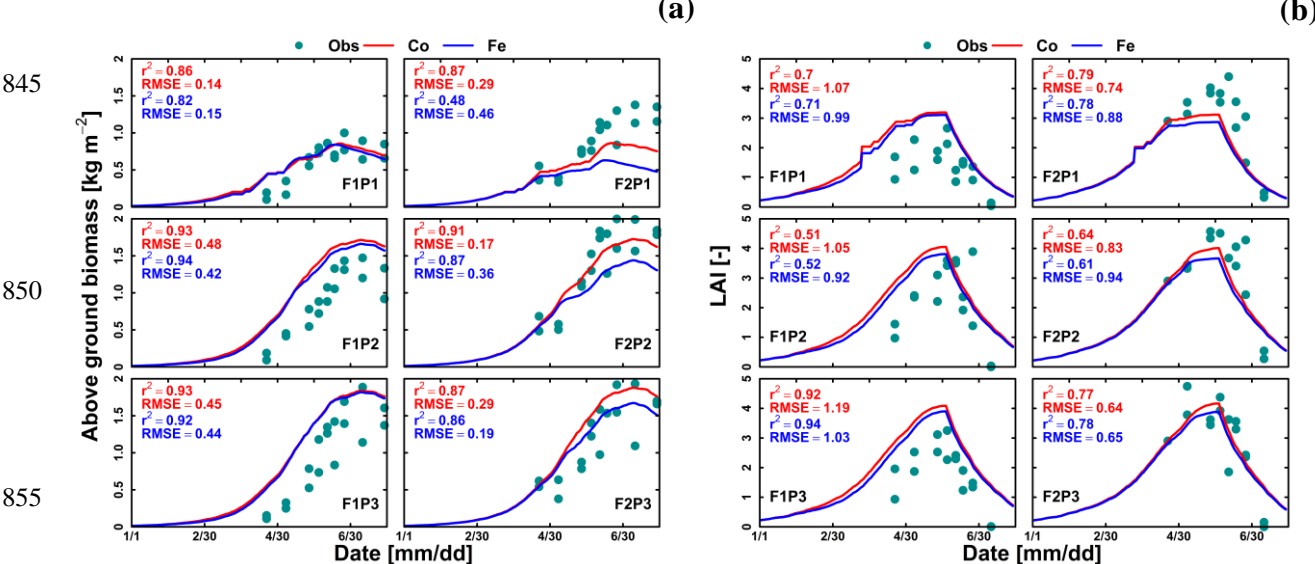

Figure 2: Comparison between observed (cyan dot) and simulated (a) above ground dry matter and (b) LAI by Couvreur (Co, solid red line), and Feddes (Fe, solid blue line) model at the sheltered (P1), rainfed (P2), and irrigated (P3) plots of the stony soil (F1) and the silty soil (F2).

Note: crop germination was on 26th October 2015, data is shown here from 1 January to harvest 23 July 2016. RMSE in (a) is kg m⁻² while RMSE in (b) is unit less.






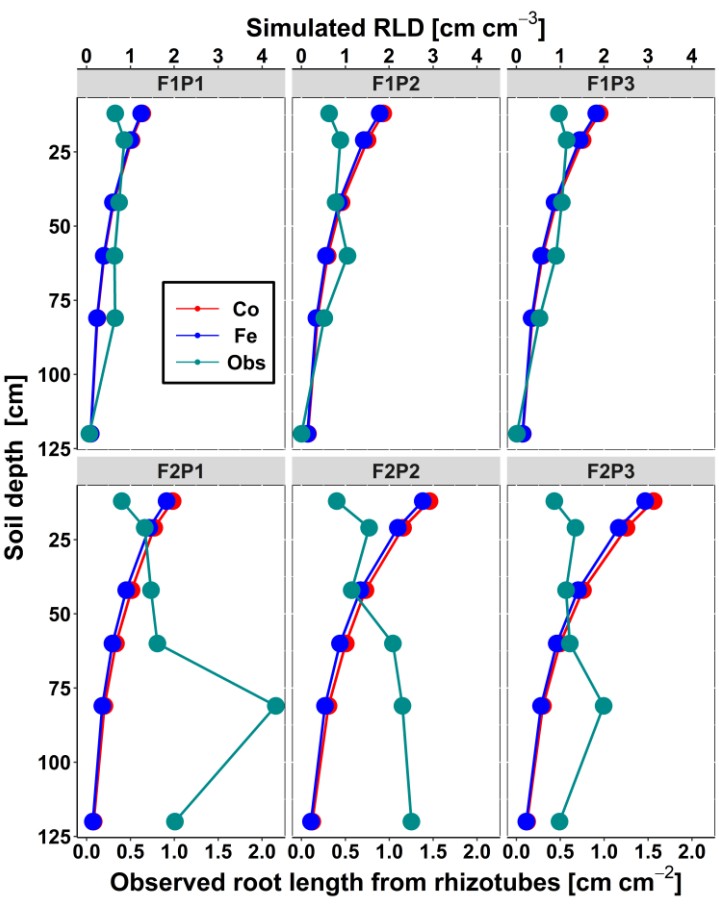

Figure 3: Comparison between observed root length from rhizotubes (cm cm$^{-2}$) (green line with dots) and simulated root length density (RLD) (cm cm$^{-3}$) from 10, 20, 40, 60, 80, and 120 cm soil depth at DOY 149 by Couvreur (Co, solid red) and Feddes (Fe, solid blue) model at the sheltered (P1) rainfed (P2), and irrigated (P3), of the stony soil (F1) and the silty soil (F2)









**Figure 4:** Daily transpiration reduction factor (fwat) (a, b) from 15 March to harvest 23 July 2016 and comparison between observed (cyan)
and simulated root water uptake (RWU) and potential transpiration simulated (c, d) by Couvreur (Co, closed red), and Feddes (Fe, closed
blue) from 30 April to 20 July 2016 model at the sheltered (P1), rainfed (P2), and irrigated (P3) plots of the stony soil (F1), and the silty soil
(F2). Note: crop germination was on 26[th] October 2015. Vertical cyan bars represent the standard deviation of the flux measurements in the
different stems. Time series of precipitation (Prec) and irrigation (Irri) are given in the panels. Vertical grey lines show days with the
measured and simulated diurnal courses of RWU, leaf water potential, and Pg as used in Figure 8.





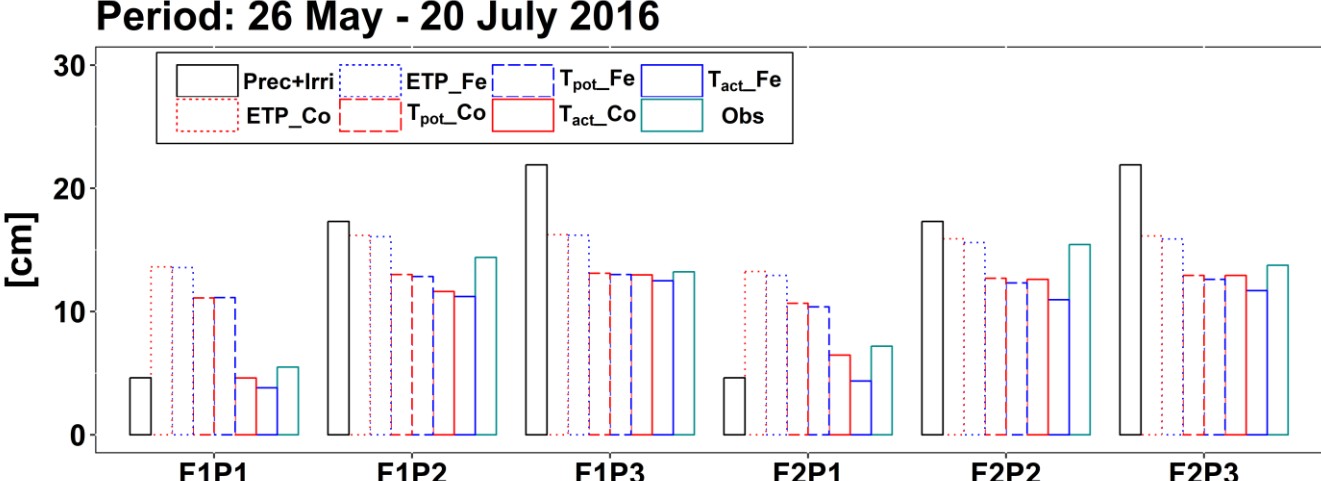

Figure 5: Cumulative precipitation and irrigation (Prec+Irri), potential evapotranspiration (ETP), potential transpiration ($T_{pot}$), actual transpiration ($T_{act}$ or RWU) simulated by Couvreur (Co) and Feddes (Fe) model, and measured transpiration by sap flow sensors (Obs) from 26 May to 20 July 2016 at the sheltered (P1), rainfed (P2), and irrigated (P3) plots of the stony soil (F1), and the silty soil (F2).






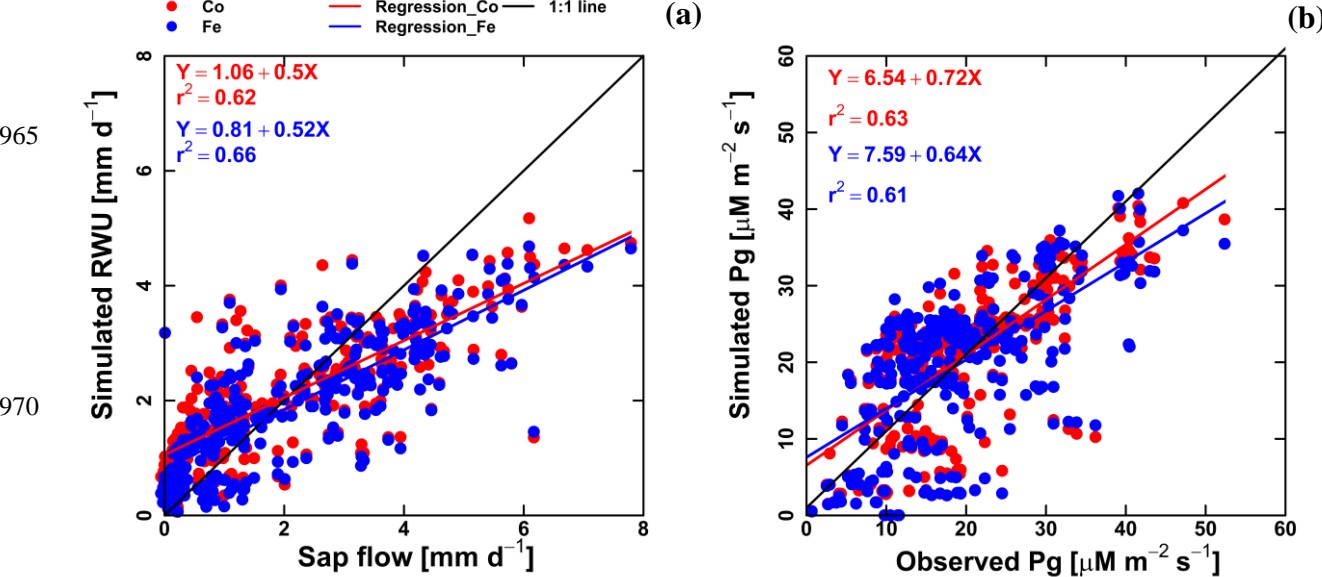

Figure 6: Correlation between observed and simulated (a) daily actual transpiration (or RWU) (b) hourly gross assimilation rate (Pg) from
Couvreur (Co, red dot), and Feddes (Fe, blue dot) models of both fields (F1 and F2). Sap flow data was from 26 May until 20 July 2017 (n
= 312). Gross assimilation rate from 08 measurement days (n = 302). RMSE in (a) is mm d$^{-1}$ while RMSE in (b) is μM m$^{-2}$ s$^{-1}$.






Figure 7: Comparison between observed (black) and simulated soil water content (SWC) by the Couvreur (Co, solid red) and Feddes RWU model (Fe, solid blue) at six soil depths in at the sheltered (P1), rainfed (P2), and irrigated (P3) plots of (a) the stony soil (F1) (b) the silty soil (F2) from 15 March to 23 July 2016. Time series of precipitation (Prec) and irrigation (Irri) are given in the panels above the SWC measurement. Triangle symbols in the sheltered plots (F1P3 and F2P3) indicate the sheltered events.




Figure 8: Diurnal courses of 4 selected measurement days: 20 April, 26 May, 9 June, and 20 June 2016 with observed (green dot) and simulated values from the Couvreur model (Co, solid red), and Feddes (Fe, solid blue) for (a & b) actual transpiration (RWU), (c & d) leaf water pressure head, and (e & f) gross assimilation rate at the sheltered plot (P1) of the stony soil (F1) and the silty soil (F2). Diurnal course

of global radiation (Rs) of the corresponding dates are given in the top panel. Sap flow sensors were installed on 26 May 2016 at 9 AM and 5 PM for F1P1 and F2P1, respectively. The Feddes RWU model did not simulate leaf water pressure head.






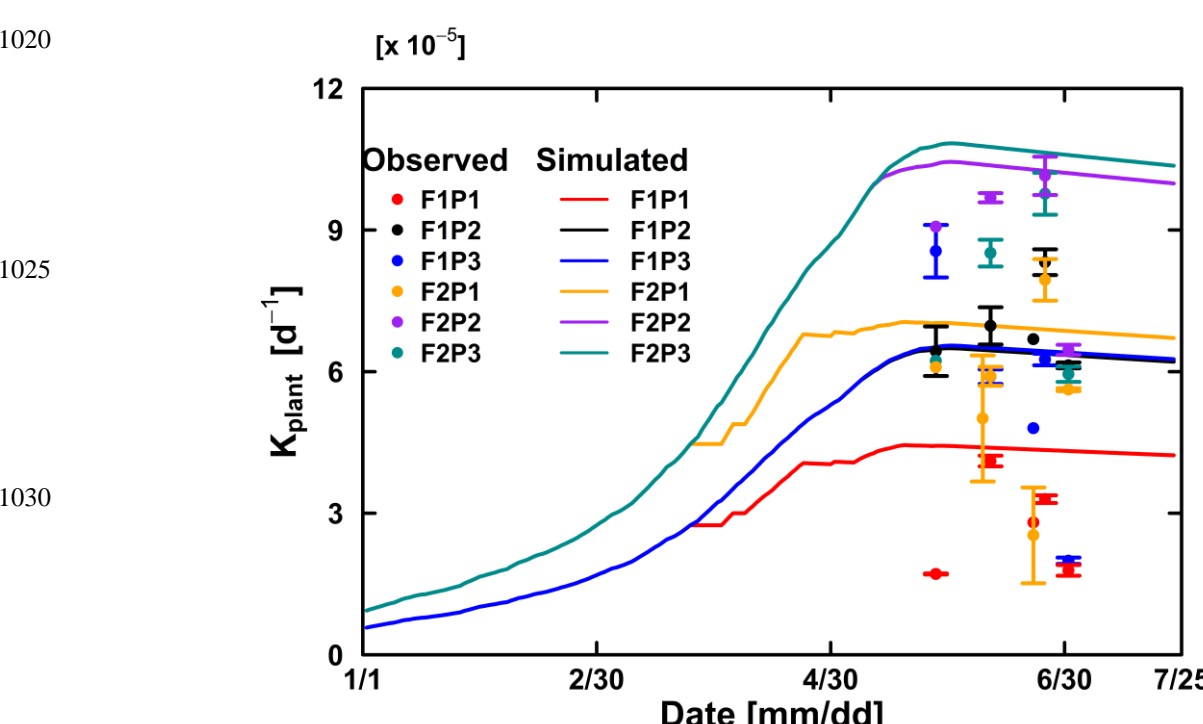

Figure 9: Comparison between observed (dot) and simulated plant hydraulic conductance (solid line) by the Couvreur (Co) model in the sheltered (P1), rainfed (P2), and irrigated (P3) plots of the stony soil (F1) and the silty soil (F2). The vertical bars represent the standard deviation of 6 hourly plant hydraulic conductance values at around midday (11 AM to 4 PM) in the measurement day. Note: crop germination was on 26th October 2015, data is showed here from 1 January 2016 to harvest 23 July 2016. Blue line was overlapped by the black line





Figure 10: Relative changes of simulated (Co model) above ground biomass at harvest (a, c, e, and g) and cumulative RWU (b, d, f, and h) (from 15 March to harvest 23 July 2016) with the changing $K_{rs, normalized}$, specific weights of seminal and lateral root, and leaf pressure head threshold ($\Psi_{threshold}$) in the sheltered (P1), rainfed (P2), and irrigated (P3) plots of the stony soil (F1) and the silty soil (F2). Vertical lines in (a) and (b) indicates the original value $K_{rs, normalized} = 0.2554 \cdot 10^{-5}$ ($d^{-1}$/cm $cm^{-2}$) while (g) and (h) indicates the $\Psi_{threshold} = -200$ m.

1080





Table *1*: Quantitative and statistical measures of the comparison between two modelling approaches and the observed data for the 3 water treatments and 2 soil types. RMSE: root mean square error; $r^2$: correlation coefficient; I: agreement index; n samples: number of sample. Couvreur RWU model (Co) and Feddes RWU model (Fe).

| Variables | Statistical indexes | Co | Fe |
|---|---|---|---|
| Daily RWU (mm d$^{-1}$) | RMSE | 1.15 | 1.13 |
| | $r^2$ | 0.62 | 0.66 |
| | I | 0.84 | 0.85 |
| | n samples | 312 | 312 |
| Biomass (g m$^{-2}$) | RMSE | 303 | 336 |
| | $r^2$ | 0.91 | 0.86 |
| | I | 0.84 | 0.81 |
| | n samples | 54 | 54 |
| LAI (-) | RMSE | 0.92 | 0.90 |
| | $r^2$ | 0.76 | 0.75 |
| | I | 0.77 | 0.77 |
| | n samples | 54 | 54 |
| Gross assimilation rate ($\mu$M m$^{-2}$ s$^{-1}$) | RMSE | 6.34 | 7.26 |
| | $r^2$ | 0.63 | 0.61 |
| | I | 0.86 | 0.83 |
| | n samples | 302 | 302 |

1085

1090