# Peer review of "Comparison of root water uptake models in simulating CO2 and H2O fluxes and growth of wheat"

_Hydrology and Earth System Sciences, 2020_

## Referee Comment (RC1) · Gaochao Cai (Referee) · 24 May 2020

In this manuscript, the authors compared the difference in water and gas fluxes for three different water treatments and two soil types using coupled water balance - root water uptake - root growth - crop growth and assimilation models. The simulations were compared with and without considering dynamic plant hydraulics using literature data, and were evaluated by sap flow measurements. It's not usual that crop growth, root growth, assimilation rate, and water uptake models were coupled for modeling water and gas fluxes. The manuscript included a lot of work, especially on the model coupling. The limitations of the coupled models were also discussed. In principle, this manuscript is well-written and the story is interesting. I have a few questions and suggestions (minor revision) that I offer in the spirit of improving clarity and message.

[Figure]

(Note: please check the attached supplement in case the comments below are not reader-friendly)

Abstract: there were several models mentioned in the abstract, which makes readers confusing for the first impression. To make it clear, I suggest to use the name that the author described in section 2.3.5: HILLFLOW–Couvreur–SLIMROOT–LINTULCC2 for the newly coupled model and HILLFLOW–Feddes–SLIMROOT–LINTULCC2 for the commonly approach at line 18.

Introduction: The author wanted to simulate the water and gas fluxes for different water and soil conditions considering dynamic plant hydraulics. The importance of dynamic plant hydraulics was well presented. Dynamic plant hydraulics related to root growth so root development was needed. The author used SLIMROOT but did not mention and explain why it was chosen in the introduction.

Methods and materials: This manuscript described five different models and each of them has different parameters (input and output). It's good that all the parameters and related values were listed in the supplementary materials. These five models were coupled in two different ways and the input or output of the models were used from each other. I suggest the author draw a diagram or flowchart to describe the connection between the models, which will definitely help readers to understand better. For the root growth model, soil water content and soil temperature were needed for the simulations. It seems that the author used simulated results from two separate models. Why did not the authors use the measured data from the soil sensors for the root growth simulations? Stomatal conductance (gs) could also be used for explaining the variation of the transpiration, especially for dry conditions. The reduction in gs shows water stress. Since these data were available (Appendix A) the variation of gs and fwat could be related somehow. The author could show and discuss it in the results and discussion part.

Results and discussion: line 479 and this paragraph were a little bit off, especially the

comparison between the modern and old cultivars. This part could be either skipped or connected with a better explanation.

Conclusion: the aim of the study, the drawback of the models, and further investigations were well presented. The three objectives of the study were posted at the end of the introduction and they were tested in the manuscript but not all of them were mentioned in the conclusion part. Normally, answers should be given in the end.

Here are some other detailed comments and suggestion: Line 10: the sentence is really long, please rephrase. Line 22: LAI is not defined before, please give the full name, leaf area index Line 30: 'promissing' – 'promising' Line 39: move (RWU) to the former 'root water uptake'. Please also check the usage of 'RWU' and 'root water uptake' in the text below. Once it is described, the abbreviation should be used afterwards. Line 47: 'in an indirect manner' – 'indirectly' Line 50: 'models of root water uptake' – 'RWU models' Line 65: delete 'shoot' Line 86: missing the 'period' symbol Line 106: delete 'soil property' since the soils have been described before Line 110: 'side' – 'sides' Line 112: 'was' – 'were' Line 115: 'rain-fed' – 'rainfed' and also check it in the text below Line 119: . . . sap flow was calculated . . . Line 123: '8pm' – '8 pm' Line 130 and 131: '6' – 'six' Line 132: use am (pm) or AM (PM) in the whole text Line 150: 'above ground' – 'aboveground', and also check it in the text below Line 151: the detailed measurements of biomass, especially the different organs, were described but not used later. This part could be skipped Line 162: . . . model of Farquhar and Caemmer (1982) Line 165: For the sake of . . . Line 167: check the format of the citations in the bracket Line 171: give the full name of LAI Line 190: keep 'Hillflow1D' and 'HILLFLOW 1D' the same in the text Line 229: 'fwat' – 'fwat(subscript)' and also in Figure 4 Line 323-324: Not clear. It is better to have two different colors or symbols to differentiate the two samples. Line 325: do you use the mean r2 of the six plots? If so, you need to mention and also re-calculate them. It seems that 0.91 and other values are not the mean of the six values. Line 343: use 'minirhizotube' or 'rhizotube' in the text and in the caption of the figures Line 363: . . . show the simulated . . ., by the

. . .' – '. . . show . . ., simulated by . . .' Line 407: 'Pg' is not defined Line 463: 'increases' – 'increase' Line 477: 'is' – 'are' Figure 2, Line 879: 'green' – 'cyan' (used in Figure 2 and 4) Figure 4: make the size of the four subplots (a, b, c, d) the same for better comparison Figure 4, line 934: Pg? Please give the full name Appendix F: bar plot will be better for the comparison

Please also note the supplement to this comment:
https://www.hydrol-earth-syst-sci-discuss.net/hess-2020-180/hess-2020-180-RC1-supplement.pdf

---

## Referee Comment (RC2) · Anonymous Referee #2 · 7 Jul 2020

General Comments

This study compares two crop and root growth models with measured data that were previously published. The models differ mainly in their representation of root water uptake. One uses the standard approach (Feddes), the other considers the flow process in the roots (but not to the roots) and is hence more mechanistic. The paper is fairly well written. It is a valuable contribution to soil/crop science, but would benefit from a few extensions and corrections, as outlined below.

It is a severe shortcoming of the measurements that the field plots do not involve replications. Replicates would be very desirable to enable assessing the variability, but their omission should not prevent the manuscript from being published. It should be frankly stated that and why there were no replicates (too expensive?).The experiment should be described better (e.g., the plot size is not given).

In model-measurement comparisons, it is good practice to present the measured data with standard deviations or errors and the modeled data as lines; if uncertainty is considered, with uncertainty bands. This is not always the case here (Figure 3 no, figure 4 yes; why?)

The paper should give a few more details about the calibration of the soil-crop model. The role of Penman's ETP should be discussed.

The Conclusions section should be improved. The first paragraph is misplaced. I would like to hear a bit more about the rhizosphere conductivity under drought and see the work (at least one paper) of Andrea Carminati cited in this context. Model testing is important but how could the model be improved? Is it not a severe shortcoming that the drop in root length density in the topsoil is neglected? And, similarly, the increased root growth under drought stress? How could this be represented better in the model? What models are already out there that are capable of handling such situations?

Does the fact that P1 receives less water but is exposed to the same weather situation in regard to all other weather variables (e.g., air humidity) as P2-P3 might have biased (in the sense of an artifact) the reaction of crops in the field as compared to the simulations?

Detailed comments line 41 function of correct: are lost make clear that you use the terms water potential and hydraulic potential coherently. Better, define it for the readers from different fields. There is a problem because traditionally for plant scientists water potential does not contain the gravitational component, for soil scientists it does. What is root zone water potential? Is it the hydraulic or matric potential in the rhizosphere? probably not.

computation of

In both models, delete "for each model"

235, 240 in a given layer delete "sufficient"

I recommend deleting "which is based on a mechanistic description of water flow in the coupled soil-plant system," because you are her in the technical part.

delete "the"

UTC is more confusing than local time better "characterizes the difference" or "is a measure for the difference"

"are uncertain"

seminal roots units missing threshold (index)

better reverse: kplant explicitly simulated by. . .

we present and discuss the results of the sensitivity analysis in fair agreement (at best)

Grammar: this should not be emphasized too much Content: This cannot be emphasized too much because it shows a clear and important shortcoming of your modeling approach and gives a point of leverage for the next step of improvement.

better: transpiration rates simulated by the Fe/Co model or simply transpiration rates by the Fe/Co model less dry

"from" end of May

I do not understand how you define adequate. I would rather write fair.

Pg is not defined. For the reader, it is better to write it out.

the sheltered plot with the silty soil (the field is the same - according to figure 1) if this is not true add the field borders in figure 1

comma before based delete "in the measurements" (perhaps: observed in the field)

and elsewhere see above 443

must have causes not considered in the model ("other causes that" is wrong here)

The sensitivity analysis is, frankly speaking, a bit boring (sorry). It destroys the flow of the paper and feels like a "lost part". The reader should be left off the hook after Figure 9, but (recommendation) after a better discussion of what he or she can learn from all that.

are lower than those of old cultivars (not were)

indicates that potential more mechanistic, then you can drop the quotation marks no comma

How were xj and wj determined? Should it not read LAI(xj)?

"thus there was no Gaussian integration over time degree" - this cannot be understood better write "grass reference evapotranspiration (FAO, give the reference)"

reference needed

How were surface resistance and aerodynamic resistance calculated?

verb missing matric potential, not matrix potential matric potential head should have a unit, here m

Please make clear in the whole paper if you talk about the soil matric potential or soil hydraulic potential. Otherwise it is confusing. Here, for example, I feel that you mean soil matric potential. Actually I would avoid using the expression soil water potential.

Figures All figure captions should be formulated more carefully and with more empathy for the reader.

Figure 1 Indicate what kind of rock.

Figure 7 You should try to explain the systematic deviation in the deeper soil layers. 998: The should be better described in the text.

Figure 6,8 Rephrase the confusing caption. You should start with a statement about what the reader can see. Include the top graphs in the enumeration. Psi_leaf and Pg should be defined in the caption (as RWU).

Figure 10 should be deleted

---

## Author Comment (AC1) · 27 Jul 2020

**Authors' response to interactive comment of the Referee #1**

**Black text: Referee comment**

**Blue text: Authors' response**

We thank the reviewer for the valuable comments and suggestions to improve our contribution. We provide point-by-point reply below.

**Abstract:** there were several models mentioned in the abstract, which makes readers confusing for the first impression. To make it clear, I suggest to use the name that the author described in section 2.3.5: HILLFLOW–Couvreur–SLIMROOT–LINTULCC2 for the newly coupled model and HILLFLOW–Feddes–SLIMROOT–LINTULCC2 for the commonly approach at line 18

The name of the coupled model has changed in the abstract that it is consistent with section 2.3.5.

**Introduction:** The author wanted to simulate the water and gas fluxes for different water and soil conditions considering dynamic plant hydraulics. The importance of dynamic plant hydraulics was well presented. Dynamic plant hydraulics related to root growth so root development was needed. The author used SLIMROOT but did not mention and explain why it was chosen in the introduction.

The coupling root-shoot model required a use of root growth model. For a valid comparison, it is necessary to couple both RWU models and water balance model with the same root growth model. From this point, root growth model with dynamic root growth (over time and soil layers) was chosen to inform the root depth, root length, and root length density (or normalized root length density) for Feddes and Couvreur RWU approach. Thus, the authors thought that it might not be necessary to mention and answer why SLIMROOT was employed right away from the Introduction section. However, the SLIMROOT root growth model was described in detail in the section 2.3.2.

**Methods and Materials:**

This manuscript described five different models and each of them has different parameters (input and output). It's good that all the parameters and related values were listed in the supplementary materials. These five models were coupled in two different ways and the input or output of the models were used from each other. I suggest the author draw a diagram or flowchart to describe the connection between the models, which will definitely help readers to understand better.

The modelling couple was the same for both RWU approaches. The coupling configuration namely linkages of shoot growth, root growth, water balance, and root water uptake model was the identical to ensure the analysis and comparison of two different RWU models. As suggestion of referee's comments, a diagram was added (see Figure 2). A short paragraph was added to describe shortly the Figure 2.

"For a certain hourly time step $\Delta t_i = t_i - t_{i-1}$, different modules were solved in the following sequence. First, LINTULCC2 was used with a water stress factor fwat =1 to calculate the leaf and canopy resistance, and the potential transpiration rate. $T_{pot}$ was then used in HILLFLOW 1D to calculate the soil water pressure head changes, water content changes, the actual transpiration, and fwat during the time step. LINTULCC2 was then run again using the fwat. The leaf conductance and assimilation rate were calculated. For the next time step, the same loop was run and hourly assimilation was accumulated to a daily value. Daily assimilation rates were used in modules that run with a daily time step. For instance, modules of LINTLCC2 that calculate assimilate partitioning which is used to calculate shoot (LAI) development and passed to SLIMROOT to simulate root development (Fig. 2)."

[Figure]

Figure 2: Description of the coupled root: shoot models in the study. The orange arrow indicates feedbacks from the hourly simulations to daily simulation while the grey arrow indicates feedbacks from the daily simulations to the hourly simulations. The dashed black arrows denote the weather input and parameters to the subroutines. The continuous black arrows indicate the links amongst the modelling subroutines.

For the root growth model, soil water content and soil temperature were needed for the simulations. It seems that the author used simulated results from two separate models. Why did not the authors use the measured data from the soil sensors for the root growth simulations?

The authors can used directly the measured root growth which was collected weekly by rhizotubes for the daily simulation by assuming root growth will not change much within one week (as similar to the work from Cai et al., 2017; Cai et al., 2018). In addition, the measured soil water content and soil temperature can be used as the "forced input" for the root growth simulation. We used the "forward modelling approach" which weather data, soil, and crop parameters are the input for two coupled models. Soil water content is simulated by water balance model HILLFLOW 1D. The soil temperature is calculated by the subroutine in the root growth model (STMPsim, Williams and Izaurrable, 2006) (see the new diagram Figure 2). The output data for instance here the performance of root growth simulation was evaluated directly with the observed root data while the simulated soil water content was compared with the measured soil water content data.

Stomatal conductance (gs) could also be used for explaining the variation of the transpiration, especially for dry conditions. The reduction in gs shows water stress. Since these data were available (Appendix A) the variation of gs and fwat could be related somehow. The author could show and discuss it in the results and discussion part.

The leaf water potential is surrogate of stomatal regulation our study. By showing simulated leaf water potential, transpiration, gross assimilation in comparison to the measure data (Fig. 8), our work has showed that the model is able to simulate water stress effects on leaf water potential and gas exchange in different measured days (for instance Couvreur model). As suggested by the referee, the simulated stomatal conductance to water vapor (here from sunlit leaves) was compared with the measurement from 3-4 upmost fully developed leaves. Two sentences were added to describe the measurement of leaf gas exchange together with leaf water potential and one sentence was add for the simulation results of stomatal conductance in comparison with the measured ones.

Results and discussion: line 479 and this paragraph were a little bit off, especially the comparison between the modern and old cultivars. This part could be either skipped or connected with a better explanation.

This paragraph was removed and improved (see also reply to the comments with sensitivity analysis section from referee # 2)

Conclusion: the aim of the study, drawback of the models, and further investigations were well presented. The three objectives of the study were posted at the end of the introduction and they were tested in the manuscript but not all of them were mentioned in the conclusion part. Normally, answers should be given in the end.

The Conclusion part was revised (together with comments from referee #2)

Line 10: the sentence is really long, please rephrase.

This was rephrased and shortened down.

Line 22: LAI is not defined before, please give the full name, leaf area index

It was given full name.

Line 30: 'promissing' – 'promising'

It was corrected.

Line 39: move (RWU) to the former 'root water uptake'. Please also check the usage of 'RWU' and 'root water uptake' in the text below. Once it is described, the abbreviation should be used afterwards.

 It was moved.

Line 47: 'in an indirect manner' – 'indirectly'

 It was corrected.

Line 50: 'models of root water uptake' – 'RWU models'

It was corrected.

Line 65: delete 'shoot'

It was deleted.

Line 86: missing the 'period' symbol

It was added.

Line 106: delete 'soil property' since the soils have been described before

It was deleted.

Line 110: 'side' – 'sides'

It was corrected.

Line 112: 'was' – 'were'

It was corrected.

Line 115: 'rain-fed' – 'rainfed' and also check it in the text below

It was corrected and made consistently for the next paragraphs.

Line 119: … sap flow was calculated …

It was corrected.

Line 123: '8pm' – '8 pm'

It was corrected.

Line 130 and 131: '6' – 'six'

It was corrected.

Line 132: use am (pm) or AM (PM) in the whole text

These will be made consistently in the whole text.

Line 150: 'above ground' – 'aboveground', and also check it in the text below

These will be changed consistently in the whole text.

Line 151: the detailed measurements of biomass, especially the different organs, were described but not used later. This part could be skipped

It was kept like this because the separated organs need to be measured then the total aboveground biomass is determined from the sum of different organs.

Line 162: … model of Farquhar and Caemmer (1982)

It was corrected.

Line 165: For the sake of …

It was corrected.

Line 167: check the format of the citations in the bracket

It was corrected with the right citation format.

Line 171: give the full name of LAI

The full name of LAI was added

Line 190: keep 'Hillflow1D' and 'HILLFLOW 1D' the same in the text

It was corrected and made consistently in the text.

Line 229: 'fwat' – 'fwat' and also in Figure 4

It was corrected.

Line 323-324: Not clear. It is better to have two different colors or symbols to differentiate the two samples.

These were two replications of biomass and LAI. It was rewritten for better understanding.

Line 325: do you use the mean r2 of the six plots? If so, you need to mention and also re-calculate them. It seems that 0.91 and other values are not the mean of the six values.

The $r^2$, RMSE, and I were calculated when all measurements from 06 plots were pulled together (they are not the mean of six values from six plots).

Line 343: use 'minirhizotube' or 'rhizotube' in the text and in the caption of the figures

The minirhizotube will be used and changed consistently in the text and the figure captions.

Line 363: … show the simulated …, by the …' – '… show …, simulated by …'

It was revised.

Line 407: 'Pg' is not defined

Pg will be defined in the text.

Line 463: 'increases' – 'increase'

It was corrected

Line 477: 'is' – 'are' Figure 2,

It was corrected.

Line 879: 'green' – 'cyan' (used in Figure 2 and 4) Figure 4: make the size of the four subplots (a, b, c, d) the same for better comparison Figure 4,

It was corrected. The size of the subplots will be similar in Figure 4.

line 934: Pg? Please give the full name

The full name was added.

Appendix F: bar plot will be better for the comparison

It was converted to bar plot.

[Figure]

Appendix F: Comparison ratio of the observed total root length from minirhizotubes to observed total root length from F1P2 (green line with squares) and ratio of simulated total root length to the simulated total root length from F1P2 on 11 July 2016 (DOY 193) from Couvreur (Co, solid red, dots), and Feddes (Fe, solid blue, triangles) model at the sheltered (P1), rainfed (P2), and irrigated (P3) plots of the stony soil (F1) and the silty soil (F2)

---

## Author Comment (AC2) · 27 Jul 2020

**Authors' response to interactive comment of the anonymous Referee #2**

Black text: Referee comment

Blue text: Authors' response

We thank the reviewer for the valuable comments and suggestions to improve our contribution. We provide point-by-point reply below.

General Comments

This study compares two crop and root growth models with measured data that were previously published. The models differ mainly in their representation of root water uptake. One uses the standard approach (Feddes), the other considers the flow process in the roots (but not to the roots) and is hence more mechanistic. The paper is fairly well written. It is a valuable contribution to soil/crop science, but would benefit from a few extensions and corrections, as outlined below.

Thank you for your comments. The measured data has not published before. However, the soil parameters and RWU parameters from two RWU models were published in the previous studies (with the same experimental set-up but in 2014). The published the soil parameters and RWU parameters from two RWU models were used in this study.

It is a severe shortcoming of the measurements that the field plots do not involve replications. Replicates would be very desirable to enable assessing the variability, but their omission should not prevent the manuscript from being published. It should be frankly stated that and why there were no replicates (too expensive?). The experiment should be described better (e.g., the plot size is not given).

Yes. We agreed that the field plots did not involve replications.

The construction and experimental designs were described in detail in Cai et al., (2016); Cai et al., (2017) and Cai et al., (2018). The authors also referred the readers for the detail explanation of field trial these papers. As suggested from the referee, we added three sentences in the section 2.1 (Location and experimental set-up) between line 93 and 94:

"Each treatment was 3.25 m wide and 7 m long. The treatments bordered each other along 7-m-long side. There were no replicates for plots due to the complex and expensive construction of underground minirhizotrone facilities".

In model-measurement comparisons, it is good practice to present the measured data with standard deviations or errors and the modeled data as lines; if uncertainty is considered, with uncertainty bands. This is not always the case here (Figure 3 no, figure 4 yes; why?)

Figure 4 showed the transpiration by two models versus the measured sap flow. The measured sap flow was achieved with 5 sensors in each plot. Thus, the variability of transpiration from different stems could be shown via error bars. The Figure 3 showed the simulated root length density versus the observed root length in 06 soil depths in different treatments. For one observation depth and one treatment, roots were counted in 120 images of 13.5 mm x 18 mm. This dataset represents a sample of the population of all possible root counts at this depth. Cai et al., (2016) estimated and analyzed the standard deviation (error) and spatial correlation of root counts along the minirhizotube. The standard deviation was small and no spatial correlation of root densities in the horizontal direction was observed for the investigated **winter wheat** crop. That is a reason that Figure 3 we did not show the standard deviation for the root measurements.

The paper should give a few more details about the calibration of the soil-crop model. The role of Penman's ETP should be discussed.

The calibration was mentioned in line 283-284. Following the suggestion from the referee, the calibration sentences was revised and is added more details.

"Before comparing these modelling approaches, we calibrated the original LINTULCC model using the data from the rainfed plots in the silty soil (F2P2). The model is firstly calibrated to make sure the model properly described the phenology. Two parameters (minimum thermal sum from sowing to anthesis and thermal sum from anthesis to maturity (°C d)) were used for phenology calibration based on information of sowing, anthesis, and maturity dates. The model was then calibrated using time series of LAI, biomass, and gross assimilation rate through the change of maximum carboxylation rate at 25 °C (VCMAX25), critical leaf area index (LAICR), and relative growth rate of leaf area during exponential growth (RGRL) parameters."

Regarding the potential evapotranspiration (ETP), our study used the Penman-Monteith equation from Allen et al., (1998) (See Equation 3, Chapter 2 in the FAO Irrigation and Drainage Paper No. 56, Allen et al., 1998). The ETP was calculated with crop canopy resistance under optimal water condition (fwat = 1) (see the Appendix B). The hourly net radiation, soil heat flux, and aerodynamic resistance were estimated based on Allen et al., (1998) (see also the replies for comments at line 559, 561, and 565). Following suggestion from the referee, some sentences were added after line 404 to describe the roles of ETP.

"The method that we used for modelling the canopy resistance used in the Penman-Monteith has been reported for both short and tall crops (Dickinson et al., 1991; Kelliher et al., 1995; Irmak & Mutiibwa, 2010; Perez et al., 2006; Katerji et al., 2011; Srivastava et al., 2018). The fair agreement of RWU to sap flow in our study indicates the proper estimate of ETP based on the crop canopy resistance (with fwat = 1) in winter wheat. The direct calculation of crop canopy resistance in our work allows to capture physiological responses of the crop (stomatal conductance) to solar radiation, temperature, and vapor pressure deficit (Eqn. A5). In addition, this approach also avoids calculating grass reference evapotranspiration based on a constant canopy resistance."

The Conclusion section should be improved. The first paragraph is misplaced. I would like to hear a bit more about the rhizosphere conductivity under drought and see the work (at least one paper) of Andrea Carminati cited in this context. Model testing is important but how could the model be improved? Is it not a severe shortcoming that the drop in root length density in the topsoil is neglected? And, similarly, the increased root growth under drought stress? How could this be represented better in the model? What models are already out there that are capable of handling such situations?

The conclusion section was revised which should answer the mentioned objectives in the Introduction (see comments from Referee # 1) and will be added some suggestions from Referee #2. The first paragraph is shortened down to conclude for the first objective. The second graph concluded the model ability in simulating plant hydraulic conductance (which is for the second objective) with more insights of rhizosphere conductance and citation from Andrea Carminati. The third paragraph concluded for the results from sensitivity analysis (third objective). The last three small paragraphs mentioned on the model limitations and outlooks. Some root growth modelling approaches will be added that we can lean and improve the models.

[revised manuscript text omitted]

Does the fact that P1 receives less water but is exposed to the same weather situation in regard to all other weather variables (e.g., air humidity) as P2-P3 might have biased (in the sense of an artifact) the reaction of crops in the field as compared to the simulations?

We agreed and understood that in the field, the rainout shelters might influence the P1 itself and nearby P2 and P3 plots with regard to air circulation (and thus air humidity and canopy temperature). We also expected that there could be a difference with regards to microclimate conditions and crop reactions amongst the plots. We tried to minimize as much as possible the effects of shelter application on climatic conditions and crop growth. The rainout shelter was used when it rains and removed directly when rain stops to minimize the effects of the shelter within the plot P1 and on plot P2 and P3. Water was collected by the shelter from P1 was drained out that did not pour on the P2.

Detailed comments line 41 function of

It was corrected correct: are lost

It was corrected make clear that you use the terms water potential and hydraulic potential coherently. Better, define it for the readers from different fields. There is a problem because traditionally for plant scientists water potential does not contain the gravitational component, for soil scientists it does. What is root zone water potential? Is it the hydraulic or matric potential in the rhizosphere? probably not.

It was corrected. One sentence was added to clarify the terms in the introduction part and used consistently for the whole paper.

computation of

It was corrected

In both models, delete "for each model"

It was corrected

235, 240 in a given layer

It was corrected delete "sufficient"

It was changed

I recommend deleting "which is based on a mechanistic description of water flow in the coupled soil-plant system," because you are her in the technical part.

It was deleted delete "the"

It was deleted

UTC is more confusing than local time

We do not understand clearly this comment. It is important to match time of weather input data and output data from the models with the time of the measured data. The conversion of the local time (CEST and CET in Germany) to UTC time aims at avoiding the time confusion since the UTC is standard time which might help the paper targets to "broader readers".

better "characterizes the difference" or "is a measure for the difference"

It was changed "characterizes the difference".

"are uncertain"

It was changed.

seminal roots

The range of value is for the lateral root. This will not be changed.

units missing

Unit was added.

threshold (index)

We agree, threshold was added.

better reverse: kplant explicitly simulated by...

This was changed.

we present and discuss the results of the sensitivity analysis

It was revised.

in fair agreement (at best)

It was revised.

Grammar: this should not be emphasized too much Content: This cannot be emphasized too much because it shows a clear and important shortcoming of your modeling approach and gives a point of leverage for the next step of improvement.

It was revised.

better: transpiration rates simulated by the Fe/Co model or simply transpiration rates by the Fe/Co model

It was revised.

less dry

It was revised.

"from" end of May

I do not understand how you define adequate. I would rather write fair.

It was revised.

Pg is not defined. For the reader, it is better to write it out.

Thank you. The Pg is defined.

the sheltered plot with the silty soil (the field is the same - according to figure 1) if this is not true add the field borders in figure 1

This is in line 433 not line 443. The two sites were in the same field (around 200 m long). There is no field borders however the two sites were situated in two different soil types. The text was revised.

comma before based

It was revised.

delete "in the measurements" (perhaps: observed in the field)

It was revised.

and elsewhere see above 443

It was revised.

must have causes not considered in the model ("other causes that" is wrong here)

It was revised.

The sensitivity analysis is, frankly speaking, a bit boring (sorry). It destroys the flow of the paper and feels like a "lost part". The reader should be left off the hook after Figure 9, but (recommendation) after a better discussion of what he or she can learn from all that.

Thank you very much for your suggestion. The couple root: shoot model with such the mechanistic RWU Couvreur model with considering two ways coupling has not been done before. This is the first study to evaluate the performance of the coupled root: shoot model. Thus, the sensitivity analysis is strongly necessary because of several reasons (i) to understand the roles of important parameters of Couvreur model itself (critical leaf water pressure head and plant hydraulic conductance) and other root hydraulic conductance parameters which are rarely available at field scale (see Cai et al. 2017 and Cai et al., 2018) or the root parameters which are often used in crop models that might contribute to plant hydraulic conductance (ii) to understand the feedbacks and effects of aboveground biomass, root growth, root system hydraulic conductance, whole plant hydraulic conductance, and leaf water pressure head threshold on RWU and biomass (iii) the feedbacks of aboveground crop growth to belowground can be only analyzed with Couvreur model (iv) by doing this analysis, the important roles of hydraulic conductance and necessity of two-ways couple can be emphasized. Thus, the authors would like to keep this section (together with Fig.10). However, the section is shortened down. Line 479 to lines 487 will be deleted (see also the replies to comments from Referee #1 on this paragraph).

are lower than those of old cultivars (not were)

It was revised.

indicates that

 It was revised.

potential

It was revised.

more mechanistic, then you can drop the quotation marks

It was revised.

no comma

This in line 499. It was revised.

How were xj and wj determined? Should it not read LAI(xj)?

The formulation was revised by adding the multiplicative symbols. An integral from [0, LAI] needs to be changed into integral over [-1, 1] before using the Gauss –Legendre quadrature (Stoer and Bulirsch, 2002). The estimate of $x_j$ and $w_j$ can be derived from page 178, Chapter 3.6. "Gausian Integration methods". The $x_j$ and $w_j$ can also be found from https://en.wikipedia.org/wiki/Gaussian_quadrature.

"thus there was no Gaussian integration over time degree" - this cannot be understood

It was revised better write "grass reference evapotranspiration (FAO, give the reference)"

Please see our above reply for the role of ETP calculation with Penman-Monteith equation. This is not grass reference evapotranspiration. The potential evapotranspiration (ETP) is calculated by hourly Penman-Monteith equation from Allen et al., (1998) (See Equation 3, Chapter 2 in the FAO Irrigation and Drainage Paper No. 56, Allen et al., 1998). The ETP is calculated from non-stress crop canopy conductance (fwat = 1, see Figure 2).

reference needed

Reference was added

How were surface resistance and aerodynamic resistance calculated?

The surface resistance was corrected. The surface resistance is crop canopy resistance (Eqn. B5). The hourly aerodynamic resistance ($r_a$) is calculated as Equation 4, Chapter 2 in the FAO Irrigation and Drainage Paper No. 56, Allen et al., (1998).

verb missing matric potential, not matrix potential matric potential head should have a unit, here m Please make clear in the whole paper if you talk about the soil matric potential or soil hydraulic potential. Otherwise it is confusing. Here, for example, I feel that you mean soil matric potential. Actually I would avoid using the expression soil water potential.

This term of was changed in the paper.

Figures

All figure captions should be formulated more carefully and with more empathy for the reader.

The captions were revised.

Figure 1 Indicate what kind of rock.

The rock type was indicated.

Figure 7 You should try to explain the systematic deviation in the deeper soil layers. 998: The should be better described in the text.

The Caption was revised. Thank you for your suggestion. The systematic deviation in the deeper soil layers was explained from Line 419 to 425.

Figure 6, 8 Rephrase the confusing caption. You should start with a statement about what the reader can see. Include the top graphs in the enumeration. Psi_leaf and Pg should be defined in the caption (as RWU).

The caption was revised. The enumeration is added for the top graph. The full names of RWU, Pg, and $\psi_{leaf}$ were defined.

Figure 10 should be deleted

This Figure 10 was kept, please see reply Line 455 for the "sensitivity analysis" section

**Additional references:**

Carminati, A., Vetterlein, D., Weller, U., Vogel, H.-J., & Oswald, S. E. (2009). When Roots Lose Contact. *Vadose Zone Journal*, *8*(3), 805–809. https://doi.org/10.2136/vzj2008.0147

Dickinson, R. E., Henderson-Sellers, A., Rosenzweig, C., & Sellers, P. J. (1991). Evapotranspiration models with canopy resistance for use in climate models, a review. *Agricultural and Forest Meteorology*, *54*(2–4), 373–388. https://doi.org/10.1016/0168-1923(91)90014-H

Dunbabin, V. M., Postma, J. A., Schnepf, A., Pagès, L., Javaux, M., Wu, L., Leitner, D., Chen, Y. L., Rengel, Z., & Diggle, A. J. (2013). Modelling root-soil interactions using three-dimensional models of root growth, architecture and function. *Plant and Soil*, *372*(1–2), 93–124. https://doi.org/10.1007/s11104-013-1769-y

Irmak, S., & Mutiibwa, D. (2010). On the dynamics of canopy resistance : Generalized linear estimation and relationships with primary micrometeorological variables. *Water Resources Research*, *46*, 1–20. https://doi.org/10.1029/2009WR008484

Kage, H., Kochler, M., & Stützel, H. (2004). Root growth and dry matter partitioning of cauliflower under drought stress conditions: Measurement and simulation. *European Journal of Agronomy*, *20*(4), 379–394. https://doi.org/10.1016/S1161-0301(03)00061-3

Katerji, N., Rana, G., & Fahed, S. (2011). Parameterizing canopy resistance using mechanistic and semi-empirical estimates of hourly evapotranspiration : critical evaluation for irrigated crops in the Mediterranean. *Hydrological Processes*, *129*(August 2010), 117–129. https://doi.org/10.1002/hyp.7829

Kelliher, F. M., Leuning, R., Raupach, M. R., & Schulze, E. D. (1995). Maximum conductances for evaporation from global vegetation types. *Agricultural and Forest Meteorology*, *73*(1–2), 1–16. https://doi.org/10.1016/0168-1923(94)02178-M

Li, X., Feng, Y., & Boersma, L. (1994). Partition of photosynthates between shoot and root in spring wheat (Triticum aestivu, L.) as a function of soil water potential and root temperature. *Plant and Soil*, *164*, 43–50. https://link.springer.com/content/pdf/10.1007/BF00010109.pdf

Mboh, C. M., Srivastava, A. K., Gaiser, T., & Ewert, F. (2019). Including root architecture in a crop model improves predictions of spring wheat grain yield and above-ground biomass under water limitations. *Journal of Agronomy and Crop Science*, *205*(2), 109–128. https://doi.org/10.1111/jac.12306

Perez, P. J., Lecina, S., Castellvi, F., Mart, A., & Villalobos, F. J. (2006). A simple parameterization of bulk canopy resistance from climatic variables for estimating hourly evapotranspiration. *Hydrological Processes*, *532*(December 2003), 515–532. https://doi.org/10.1002/hyp.5919

Srivastava, R. K., Panda, R. K., Chakraborty, A., & Halder, D. (2018). Comparison of actual evapotranspiration of irrigated maize in a sub-humid region using four different canopy resistance based approaches. *Agricultural Water Management*, *202* (February), 156–165. https://doi.org/10.1016/j.agwat.2018.02.021

Stoer, Josef; Bulirsch, Roland (2002), Introduction to Numerical Analysis (3rd ed.), *Springer,* *ISBN* *978-0-387-95452-3*.

Yin, X., & Schapendonk, A. H. C. M. (2004). Simulating the partitioning of biomass and nitrogen between roots and shoot in crop and grass plants. *NJAS - Wageningen Journal of Life Sciences*, *51*(4), 407–426. https://doi.org/10.1016/S1573-5214(04)80005-8

---

## Author Response (AR2)

Dear Dr. Sprenger,

This is very good news that the manuscript (hess-2020-180) was finally accepted by the journal. I was very happy to receive your email. On behalf of colleagues, I would like to express my deep thanks to your acceptance as a handling editor and your enthusiastic work, and suggestions together with the reviewers to improve the manuscript. Also my many thanks are to the HESS publication team for their work and support to speed up the publication process.

There are some minor changes in the accepted version of manuscript.

*Line 447 The suggestion from reviewer 2 at line 447 was revised.*

*Line 656: The acknowledgment for two reviewers was added*

*Line 658: one sentence regarding publication processing charges was added.*

*Fig. 9 line 1018: "cyan, solid red, and solid blue dots denote" – "cyan dots, solid red lines, and solid blue lines denote"*

If you have further requirements or information, please kindly let me know.

We wish you all the best and stay strong during the Corona pandemic.

With kind regards

Thuy Huu Nguyen